# Pain-free resting-state functional brain connectivity predicts individual pain sensitivity

Tamas Spisak[1]*, Balint Kincses[2], Frederik Schlitt[1], Matthias Zunhammer [1], Tobias Schmidt-Wilcke[3,4],
Zsigmond T. Kincses[2] & Ulrike Bingel[1]

Individual differences in pain perception are of interest in basic and clinical research as altered pain sensitivity is both a characteristic and a risk factor for many pain conditions. It is, however, unclear how individual sensitivity to pain is reflected in the pain-free resting-state brain activity and functional connectivity. Here, we identify and validate a network pattern in the pain-free resting-state functional brain connectome that is predictive of interindividual differences in pain sensitivity. Our predictive network signature allows assessing the individual sensitivity to pain without applying any painful stimulation, as might be valuable in patients where reliable behavioural pain reports cannot be obtained. Additionally, as a direct, non-invasive readout of the supraspinal neural contribution to pain sensitivity, it may have implications for translational research and the development and assessment of analgesic treatment strategies.

[1] Department of Neurology, University Hospital Essen, Hufelandstrasse 5545147 Essen, Germany. [2] Department of Neurology, University of Szeged, Tisza Lajos krt. 113, 6725 Szeged, Hungary. [3] Institute of Clinical Neuroscience and Medical Psychology, University of Düsseldorf, Universitätsstraße 1, 40225 Düsseldorf, Germany. [4] Mauritius Therapieklinik, Strümper Str. 111, 40670 Meerbusch, Meerbusch, Germany. *email: tamas.spisak@uk-essen.de

Pain is a subjective, unpleasant sensory and emotional experience[1] that is highly variable across individuals[2,3]. Individual differences in pain perception are of key interest in clinical practice as altered pain sensitivity is both a characteristic and risk factor for many pain conditions[3–5]. In the past decades, brain imaging has revealed the richness and complexity of brain activity underlying both the acute pain experience[6] and pain sensitivity[7,8]. Still, the central nervous mechanisms determining individual differences in pain perception are poorly understood, partly because past neuroimaging research has mainly focussed on the momentary (acute or chronic) pain experience. The common practice of using pain-free episodes merely as a baseline reference makes studies inherently blind to components of brain activity that are not time-locked to painful events but still central to pain processing and perception.

Brain activity in the resting-state (i.e. in absence of any task or stimulation) is known to mirror some, if not all, task-induced activity patterns[9]. For instance, the well-known large-scale resting-state networks[10] (RSNs) strongly resemble patterns of related tasks[11]. Moreover, resting-state activity can predict behavioural performance, perceptual decisions and related neural activity[12,13]. Given the tight link between resting-state and task-induced brain activity, it is highly plausible that activity and functional connectivity during pain-free resting-state conditions reflect the individual's sensitivity to pain. Following the RSN-related terminology, we refer to this type of neural activity as the resting-state network of pain sensitivity.

This proposed pain-related resting-state network might have been captured by studies reporting that brain activity and connectivity directly preceding pain is associated with the subsequent pain experience[14–21]. Several characteristics of the resting-state fMRI signal in pain-free state are also known to be associated with the neural response to nociception and the resultant pain experience[22–24], its effect on cognitive performance[25] and its changes due to prior pain experience[26]. However, due to the small sample sizes, highly varying methodology (e.g. regarding the correction of physiological and motion artefacts) and the lack of validation in these previous studies the predictive power and clinical relevance of this kind of resting-state brain activity remains unclear to date[27].

Mapping the resting-state network of pain sensitivity and exploiting its capacity to predict various aspects of pain processing would substantially advance the field—both from a basic research and translational perspective. First, contrasting it with experimental pain responses would extend our understanding of how the subjective experience of pain emerges from brain activity. Second, investigating how the hypothesised resting-state pain sensitivity network is embedded into the broader resting-state brain activity could extend our knowledge about the complex functional architecture of the resting brain.

Finally, and most importantly, a robust, generalizable and rigorously validated prediction of pain sensitivity—based on the resting-state network of pain sensitivity—could lay the foundations for a non-invasive neuromarker of an individual's sensitivity to pain. Such a resting brain network-based biomarker could contribute to the development of a future pain biomarker composite signature[27] that could aid the assessment of an individual's risk of developing pain, and the objective characterization of pain conditions and analgesic treatment effects in experimental and clinical pain research.

Here, we investigate the capacity of pain-free resting-state functional brain connectivity to predict individual pain sensitivity (defined as a composite measure of heat, cold and mechanical pain thresholds) in a sample of a total of $N = 116$ young healthy participants. We first perform a whole-brain search for specific features of the pain-free resting-state connectome, which are predictive for individual pain sensitivity in a sub-study used only for the training and internal validation of the predictive model. Then, we perform a prospective validation of the approach in terms of predictive performance, generalisation and potential confounders in two independent sub-studies acquired at different scanning sites (external validation). Finally, we perform a reverse-mapping of the predictive model to identify the key nodes of the hypothesised network, hereinafter referred to as the signature of the Resting-state Pain sensitivity Network (abbreviated as RPN-signature).

## Results

**Functional connectivity-based prediction and multicentre validation.** Resting-state functional MRI data were obtained from a total of $N = 116$ healthy volunteers over three separate sub-studies, performed in three different imaging centres. Neuromarker development was based on intrinsic whole-brain functional brain connectivity, the degree to which resting-state brain activity in distinct neural regions is correlated over time (in the absence of any explicit task). Functional connectivity was assessed between $M = 122$ functionally defined regions (Fig. 1). Heat, cold and mechanical pain thresholds acquired according to the well-established quantitative sensory testing (QST) protocol[28] were aggregated into a composite pain-sensitivity score, as previously reported[8], to obtain a general estimate of pain sensitivity (see Supplementary Note 1 for rationale). Whole-brain resting-state functional connectivity data of study 1 ($N_1 = 35$, after exclusions) was used as the input feature-space ($P = 7503$ features per participant) to predict individual pain sensitivity scores, leading to a typical large P—small N setting.

According to these conditions, we constructed a machine-learning pipeline, consisting of feature-normalisation, feature-selection and fitting an elastic net regression model.

Model training consisted of fitting the pipeline and optimising its hyperparameters in a leave-one-participant-out cross-validation framework to improve generalisation to new data.

In Study 1, QST-based pain sensitivity values ranged from −1.45 to 1.52 with a robust range (range between the 5th and 95th percentiles) of 2.57 (arbitrary units).

In the internal validation (i.e. performance on left-out participant data) the model predicted pain sensitivity with a mean squared error of $MSE_1 = 0.32$ ($p_{MSE,1} < 0.0001$, Explained variance *Expl. Var.*$_1 = 39\%$, Pearson's $r_1 = 0.63$, $p_{r,1} < 0.0001$, Fig. 2b). Diagnostics of the model fit (learning-curve analysis, Fig. 2a) suggested that the approach reduced overfitting and that sample size was sufficient for an acceptable generalisation. The machine-learning pipeline with the optimal hyperparameters was finally fitted to the data of all participants in Study 1 and saved for further use. The model trained in Study 1 is henceforth referred to as the signature of the Resting-state Pain-sensitivity Network (or short, RPN signature).

As pre-registered on the 7 March 2018 (http://osf.io/buqt7), external validation studies (Studies 2 and 3) were performed in different imaging centres with different MRI scanners (from three different vendors) and with different research staff. The multi-centre design, together with a reasonable variability in imaging sequences introduced an inherent heterogeneity, ensuring that test samples are maximally independent and provide a realistic estimate of prediction accuracy and generalisability.

In Studies 2 and 3 ($N_2 = 37$ and $N_3 = 19$, after exclusions), QST-based pain sensitivity values ranged from −1.82 to 1.57 and from −1.2 to 0.55, with a robust range of 2.3 and 1.43, respectively. External validation (Fig. 2c, d) revealed a considerable generalisability of the predictive model: the mean squared prediction error was $MSE_2 = 0.54$ and $MSE_3 = 0.17$, respectively

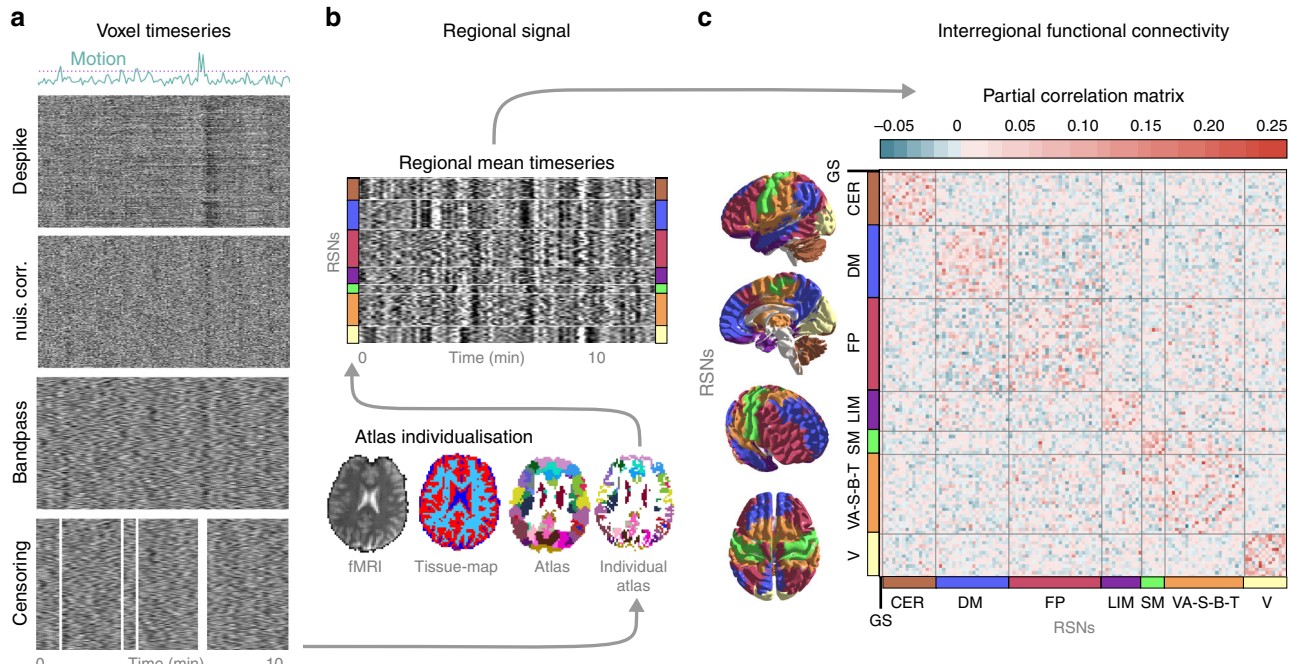

**Fig. 1 Calculating functional brain connectivity from resting-state fMRI measurements.** Raw brain images of $N = 116$ participants, in total, underwent automated artefact removal, involving despiking, nuisance regression, bandpass filtering and censoring of motion-contaminated time-frames. The effects of these procedures on the BOLD signal are exemplified on the carpet-plots (**a**, x: time, y: voxel, colour: intensity). Subsequently, a multi-stage, high-precision brain atlas individualisation was performed to obtain regional grey-matter signals for $M = 122$ functionally defined brain regions (**b**). Partial correlation between all possible region pairs was computed to asses functional connectivity and ordered based on large-scale modularity to form individual connectivity matrices. Partial correlations of all regions with the global grey-matter signal was retained to account for, but not completely discard the effect of the global signal, a component of brain activity often regarded as a confound but also related to e.g. vigilance[83]. **c** Subject-level connectivity matrices (depicted by the group-mean connectivity matrix) from Study 1 served as an input for machine-learning-based prediction of behavioural pain sensitivity.

($p_{MSE,2} = 0.02$, $p_{MSE,3} = 0.03$, *Expl. Var.₂ = 18%*, *Expl. Var.₃ = 17%*, *Pearson's $r_2 = 0.43$ and $r_3 = 0.47$, $p_{r,2} = 0.004$ and $p_{r,3} = 0.02$*). Summary statistics of pain sensitivity, in-scanner motion and demographic data are reported in Supplementary Table 1, correlations of CPT, HPT and MPT with each other and the predicted score are reported on Supplementary Note 1, Supplementary Fig. 6.

**Potential confounds and specificity to pain sensitivity.** To ensure that the RPN-signature captures the pain-related neural processing in the pain-free resting state, the potential contribution of two types of confounds has to be ruled out: (i) imaging artefacts (e.g. head motion artefacts) and (ii) demographic or behavioural variables correlated with individual pain sensitivity (e.g. age and sex are known to be slightly correlated with QST thresholds[28]).

Table 1 lists the investigated (pre-registered) confounder variables and their correlations to the predicted pain sensitivity score (together with the corresponding p-value) in all three studies. The pain sensitivity score predicted by the RPN-signature was not significantly associated with any of the confounder variables ($p > 0.05$ for all variables). Effect size was, however, considerable for sex (Study 2: $R^2 = 0.08$, $p = 0.09$), number of days from the first day of menses (Study 1: $R^2 = 0.26$, $p = 0.11$, Study 2: $R^2 = 0.11$, $p = 0.17$, Study 3: $R^2 = 0.33$, $p = 0.08$), time difference between the MRI and QST measurements (Study 2: $R^2 = 0.1$, $p = 0.06$) and with Glutamate/Glutamine levels in pain-processing regions (measured by MR spectroscopy in Study 1: $R^2 = 0.09$, $p = 0.08$). Summary statistics of confounder variables are reported in Supplementary Table 1.

Supplementary Note 1 confirms a considerable robustness of the prediction to the choice of pain threshold measures included

in the composite score (part Q3) and suggests that the prediction does not introduce any bias towards/against the investigated sensory modalities (part Q4). Supplementary Note 2 suggests that the RPN-signature displays a remarkable robustness to potential parcellation-related issues (e.g. susceptibility artefacts, drop-out effects, noise or suboptimal parcellation).

**The predictive resting-state network of the pain sensitivity.** With the applied machine-learning pipeline, non-zero regression coefficients naturally delineate the predictive sub-network. Each coefficient can be interpreted as the relative importance of the connectivity in the prediction. Positive (negative) coefficients translate to stronger interregional functional connectivity predicted higher (lower) sensitivity to pain.

The RPN-signature model, trained in Study 1, retained 21 non-zero links out of the total number of 7503 functional connections. The predictive connections are listed in Table 2 and the predictive network is depicted on the chord plot of Fig. 3b.

Almost half of the variance in the predicted pain sensitivity score is explained by the four strongest connections. The most important positive predictive connections are found between: (i) the posterior putamen (pPut) and a region including parts of the parietal operculum and the posterior superior temporal gyrus (PO/pSTG); (ii) the frontal poles (FP) and the cerebellar lobule V; and (iii) the right anterior crus II of the cerebellum and the lateral precentral gyrus (lPrCG, primary motor cortex). The only negative predictor among the top four connections was a connection between the supplementary motor cortex and the posterior part of cerebellar lobule VI. Several other interregional connections and, additionally, the global grey-matter signal was also found contribute to the predicted pain sensitivity.

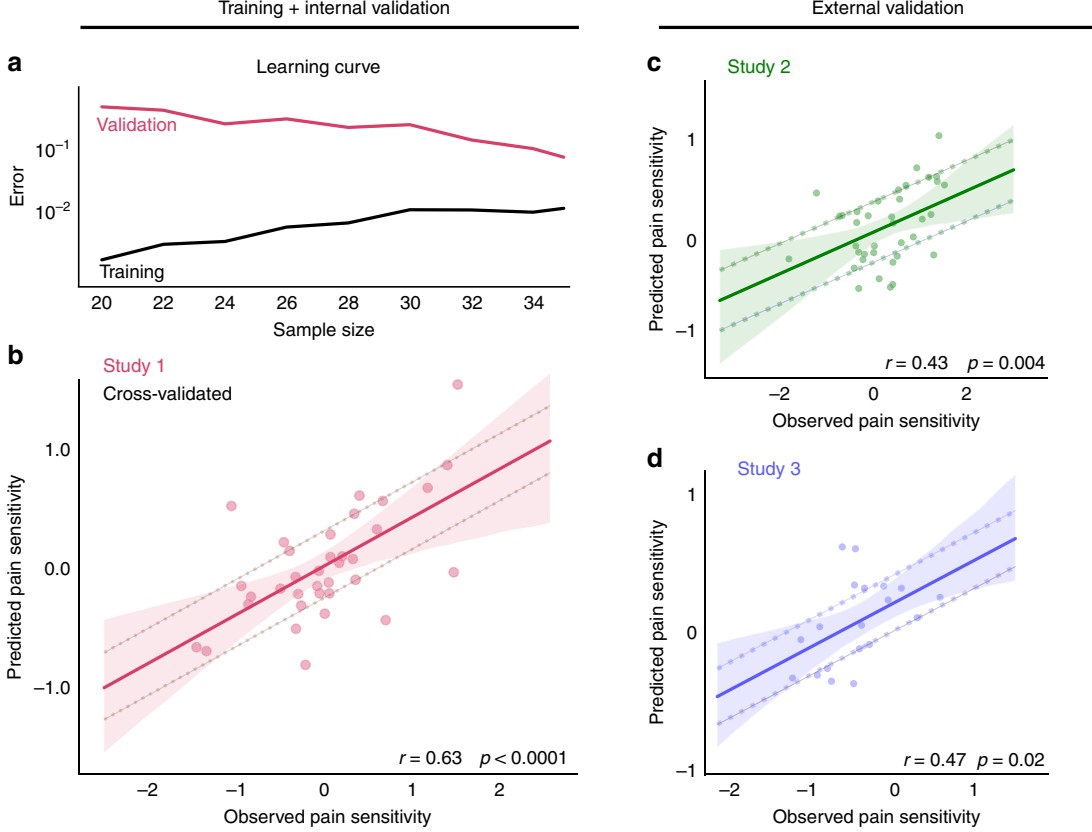

**Fig. 2 The RPN-signature predicts individual pain sensitivity based on pain-free resting-state functional brain connectivity.** The learning-curve (**a**) suggests that the size of the training sample (Study 1) was sufficient to substantially reduce overfitting and improve generalisation. Internal cross-validated prediction in the training sample (Study 1, $N = 35$, **b**) and prospective external validation in the test samples (Studies 2 and 3, **c, d**, $N = 37$ and 19, respectively) revealed considerable predictive accuracy, robustness, and multicentre generalisability of the RPN-signature. Mean absolute error (MAE) of the prediction is depicted by dashed lines. Shaded ribbons imply the 95% confidence intervals for the regression estimates, Pearson-correlation (r) of the predicted vs. observed values and the corresponding permutation-based p-value is given. Source data are provided as a Source Data file.

To simplify the overview of the spatial pattern of regions involved in the RPN-signature, we calculated the node-wise sums of predictive weights and multiplied it with the study-specific regional probability-maps. The resulting node-wise predictive strength map is displayed on Fig. 3a.

## Discussion

Here we report the RPN-signature, an objective, brain-based measure of pain sensitivity, based on functional connectivity acquired during pain-free resting-state. The applied prospective validation procedure establishes solid foundations for promising basic research and translational applications. The RPN-signature is to be applied together with a fixed resting-state fMRI analysis pipeline (https://spisakt.github.io/RPN-signature) and provides the opportunity for out-of-the-box resting-state fMRI-based, non-invasive characterisation of pain sensitivity.

This work addresses an important gap in basic research by providing strong evidence for the association of pain-free resting-state functional brain connectivity with neural processing of painful stimuli and the corollary pain experience. The identified functional network pattern provides novel insights into this—commonly unaccounted—component of resting-state brain activity and substantially advances our understanding of the neural mechanisms underlying an individual's pain sensitivity. We used a pre-registered, multicentre design and deployed a substantial sample size to perform a rigorous prospective validation of our predictive model.

Therefore, the RPN-signature may serve as an objective neuromarker of interindividual differences and alterations in pain sensitivity[3–5].

It is important to distinguish our study of the pain-free resting-state from other predictive efforts in pain research, like patient-control classification studies[29] (but also pain decoding[30]), which examine brain activity in experimental or chronic pain conditions, i.e. in the presence of painful experience. Note, that in studies of chronic pain, the terminology "resting-state" usually refers to the lack of explicit experimental pain stimuli in the data acquisition paradigm, but not to the lack of ongoing spontaneous pain experience.

In contrast, the RPN-signature is based on brain activity measured in the absence of any ongoing painful experience (which we refer to as pain-free resting-state). Therefore, it introduces a conceptually new modality for future efforts of building a composite pain biomarker[27].

The RPN-signature predicts a considerable amount of the variance in individual pain sensitivity (39% with internal validation and 18–19% with external validation, Fig. 2) which, according to Cohen's recommendations, can be considered as being in the medium-to-large range. The mean squared error (MSE) of the prediction was 0.54 and 0.17 in the external validation studies (and 0.32 with internal validation). Interpreting the magnitude of the error in comparison to the min-max-range (3.39) and the inter-quartile-range (1.04) of the observed pain sensitivity values strongly suggests that the predictive power of the RPN-signature is clinically relevant in the context of chronic

**Table 1 Confounder analysis: The RPN signature-response is specific to pain sensitivity.**

| MRI confounds | median FD | | Max FD | | % scrubbed | | BP sys. | | BP dias. | | MRI-QST dif. | |
|---|---|---|---|---|---|---|---|---|---|---|---|---|
| | R² | p | R² | p | R² | p | R² | p | R² | p | R² | p |
| Study 1 | 0.014 | 0.506 | 0.004 | 0.705 | 0.008 | 0.613 | N/A | N/A | N/A | N/A | - | - |
| Study 2 | 0.026 | 0.339 | 0.030 | 0.304 | 0.021 | 0.388 | 0.034 | 0.289 | 0.026 | 0.352 | **0.099** | **0.058** |
| Study 3 | 0.006 | 0.745 | 0.025 | 0.517 | 0.057 | 0.324 | 0.000 | 0.943 | 0.000 | 0.951 | 0.001 | 0.880 |

| Demography | Age | | Sex | | BMI | | mens. day | | alcohol | | education | |
|---|---|---|---|---|---|---|---|---|---|---|---|---|
| | R² | p | R² | p | R² | p | R² | p | R² | p | R² | p |
| Study 1 | 0.010 | 0.572 | 0.000 | 0.943 | 0.012 | 0.511 | **0.26** | 0.107 | 0.027 | 0.343 | 0.012 | 0.537 |
| Study 2 | 0.001 | 0.840 | 0.081 | **0.089** | 0.010 | 0.555 | **0.11** | 0.173 | 0.000 | 0.913 | 0.038 | 0.252 |
| Study 3 | 0.080 | 0.240 | 0.010 | 0.677 | 0.000 | 0.955 | **0.33** | **0.078** | N/A | N/A | N/A | N/A |

| Psychometrics | Pain catas. | | Pain sensitiv. | | Depression | | Anxiety state | | Sleep quality | | Stress | |
|---|---|---|---|---|---|---|---|---|---|---|---|---|
| | R² | p | R² | p | R² | p | R² | p | R² | p | R² | p |
| Study 1 | 0.000 | 0.958 | 0.011 | 0.551 | 0.002 | 0.789 | 0.002 | 0.789 | N/A | N/A | N/A | N/A |
| Study 2 | 0.062 | 0.136 | 0.017 | 0.437 | 0.021 | 0.390 | 0.021 | 0.390 | 0.008 | 0.600 | 0.003 | 0.730 |
| Study 3 | 0.012 | 0.657 | 0.005 | 0.773 | **0.094** | 0.216 | **0.094** | 0.216 | 0.029 | 0.547 | **0.127** | 0.134 |

| QST and MRS | CDT | | WDT | | MDT | | T50 | | GABA | | GLX | |
|---|---|---|---|---|---|---|---|---|---|---|---|---|
| | R² | p | R² | p | R² | p | R² | p | R² | p | R² | p |
| Study 1 | 0.002 | 0.791 | 0.002 | 0.786 | N/A | N/A | 0.039 | 0.255 | 0.048 | 0.205 | **0.089** | **0.081** |
| Study 2 | 0.014 | 0.487 | 0.000 | 0.957 | 0.041 | 0.230 | N/A | N/A | N/A | N/A | N/A | N/A |
| Study 3 | 0.019 | 0.579 | 0.055 | 0.334 | 0.027 | 0.502 | N/A | N/A | N/A | N/A | N/A | N/A |

No significant association (p < 0.05) was found with any of the confounder variables. Table headings, effect sizes with R² ≥ 0.09 (medium effect size according to Cohen) and p-values less than 0.1 are denoted by bold letters. In Study 1, MRI and QST was performed on the same day, otherwise it was measured within 1–5 days. GABA and glutamine/glutamate levels were measured by MR spectroscopy in regions of the pain matrix[8]. BP is reported here as measured on the day of the MRI measurement. See Supplementary Table 3 for additional covariates (trait anxiety and BP on the day of QST measurement, all p > 0.1). Source data are provided as a Source Data file
FD framewise displacement, % scrubbed number of censored time-frames, BP blood pressure, mens. day number of days since the first day of menstruation on the MRI-day, catas. catastrophising, sensitiv. sensitivity, CDT, WDT, MDT cold, warmth and mechanical detection thresholds, T50 temperature inducing moderate pain

**Table 2 Predictive connections of the RPN signature.**

| Predictive connections between brain regions | | | | | | Weight |
|---|---|---|---|---|---|---|
| Region | RSN | idx | region | RSN | idx | |
| PO/pSTG | VAN + SN + BG + Thal | 119 | pPut | VAN + SN + BG + Thal | 25 | 0.270 |
| FP | FPN | 75 | 5 | CER | 48 | 0.245 |
| pCVI | CER | 9 | SMC | VAN + SN + BG + Thal | 28 | −0.200 |
| R aCrus2 | CER | 62 | lPrCG | SMN | 93 | 0.150 |
| dPrCG | SMN | 67 | pmVN | VN | 51 | −0.102 |
| pdlVN | VN | 43 | mVN | VN | 40 | 0.095 |
| L IPL | DMN | 114 | mean GM | mean GM | – | −0.086 |
| vCaud | VAN + SN + BG + Thal | 2 | plVN | VN | 39 | 0.085 |
| Acc | MLN | 78 | pvmVN | VN | 107 | −0.073 |
| CF | MLN | 79 | vlPrCG | SMN | 110 | −0.062 |
| 5 | CER | 48 | pdlVN | VN | 43 | −0.059 |
| pThal/Hb | VAN + SN + BG + Thal | 36 | plVN | VN | 39 | 0.058 |
| dCVI | CER | 44 | lOTG | FPN | 117 | −0.057 |
| dCiX | CER | 11 | L vMFG | FPN | 105 | −0.056 |
| R IPS | FPN | 20 | plVN | VN | 39 | −0.054 |
| avIns | VAN + SN + BG + Thal | 12 | admVN | VN | 19 | −0.044 |
| R aMFG | FPN | 58 | lPoCG | VAN + SN + BG + Thal | 102 | 0.043 |
| CrusI | CER | 84 | dPoCG | VAN + SN + BG + Thal | 116 | −0.017 |
| pgACC | DMN | 115 | mSTG | VAN + SN + BG + Thal | 88 | 0.009 |
| Precun | DMN | 103 | LOG | MLN | 109 | −0.003 |
| vThal | VAN + SN + BG + Thal | 36 | FEF | VAN + SN + BG + Thal | 113 | −0.001 |

Non-zero regression coefficients naturally delineate the predictive sub-network. Regions and corresponding large-scale resting-state network (RSN) modules are to be interpreted as in the MIST atlas (see Methods, original atlas-index is given). Predictive connections are ordered by their absolute predictive weights. Connectivity strengths associated with higher and lower sensitivity to pain are highlighted in red and blue, respectively. For bootstrapping-based 95% confidence intervals and the p-values with conditional coverage, see Supplementary Table 4
CER cerebellum, Roman numerals cerebellar lobes, GM grey matter, VAN ventral attention network, SN salience network, BG basal ganglia, Thal thalamus, Hb habenula, MLN mesolimbic network, FPN frontoparietal network, SMN somatomotor network, DMN default mode network, VN visual network, Ins insula, PO parietal operculum, SII secondary somatosensory cortex, STG superior temporal gyrus, FEF frontal eye-field, PrCG precentral gyrus, PoCG postcentral gyrus, SMC supplementary motor cortex, Put putamen, Caud caudate nucleus, Acc nucleus accumbens, LOG lateral orbital gyrus, CF collateral fissure, OTG occipitotemporal gyrus, MFG middle frontal gyrus, IPS intraparietal sulcus, pgACC perigenual anterior cingulate cortex, PrC precuneus cortex. L left, R right, a anterior, p posterior, v ventral, d dorsal, l lateral, m medial

pain[31] and renders the RPN signature deployable in numerous applications. Here, we discuss three aspects of evaluating the relevance of the achieved predictive performance.

First, the prediction accuracy we report is similar to those in previous resting-state fMRI studies of other target variables (see e.g. refs. [32–35]). However, the majority of previously used fMRI-based predictive models were only internally validated (i.e. the same dataset was used for model training and validation) whereas our validation is based on two independently acquired samples at different scanning sites.

Second, our study is based on a large sample size ($N = 116$) and the external validation featured larger heterogeneity regarding methodology, infrastructure, research personnel and a 1–5-day delay between the QST and the MRI measurements (which were at the same day in the training dataset, see Supplementary Table 1 for further details). Our study thus overcomes recent concerns about common methodological pitfalls of neuroimaging based predictors[36]. While prediction accuracy estimates could likely be higher with a stricter standardisation of the research protocols, the above reported estimates are expected to robustly generalise to a wider variety of resting-state fMRI datasets.

Third, we believe that relying solely on the QST-based predictive accuracy might lead to underestimating the utility of the RPN-signature. While Quantitative Sensory Testing is the gold standard approach to assess pain sensitivity, it is a measure of subjective experience, shaped by peripheral, spinal and supraspinal processes convolved with perceptual and behavioural error components. The RPN-signature, on the other hand captures signal of supraspinal neural origin only. Due to this difference, prediction accuracy estimates based on the multifaceted QST-based observations should serve only as a lower bound when judging the utility of the RPN-signature as a proxy for measuring the supraspinal neural component of the interindividual variability of pain sensitivity.

Neither the investigated imaging artefacts nor the observed demographic or behavioural variables were significantly correlated with the predicted pain sensitivity values. In sum, our analysis strongly suggests that the predictive power of the RPN-signature is (i) based on signal of neural origin, (ii) is specific to pain sensitivity and (iii) is not driven by the general sensitivity to somatosensory stimuli or pain-related psychological variables such as anxiety, depression or sleep quality.

QST pain thresholds are known to differ between sexes and in different phases of the menstrual cycle. Their moderate (but statistically not significant) correlations with the predicted pain sensitivity score suggest that the RPN-signature partially captures the neural correlates of these effects. The weak ($R^2 = 0.09$, $p = 0.08$) correlation with Glutamate/Glutamine levels in pain-processing regions suggest that the RPN-signature also captures the previously reported[8] neurotransmitter-level-dependence of individual pain sensitivity. The pain sensitivity scores, predicted by the RPN-signature seem to be also slightly associated to the delay (days) between the MRI and QST measurements, which suggests that pain sensitivity and its resting-state neural correlates are subject to dynamic changes within the scale of days.

As HPT, CPT and MPT are mediated by partially different sensory pathways[24,28,37], it is interesting to evaluate how the composite score of pain sensitivity, as predicted by the RPN-signature, relates to the single pain thresholds (pain modalities). The observed moderate internal consistency and the specific correlation structure across the distinct pain thresholds have experiment-specific and neurobiological interpretations (see Supplementary Note 1 for a detailed discussion). Further, our Supplementary Note 1 corroborated previous results[38] showing that a shared component of pain processing does shape pain thresholds in all three investigated pain modalities. We found that this modality-independent component is captured by both the RPN-score and the composite pain sensitivity score used as

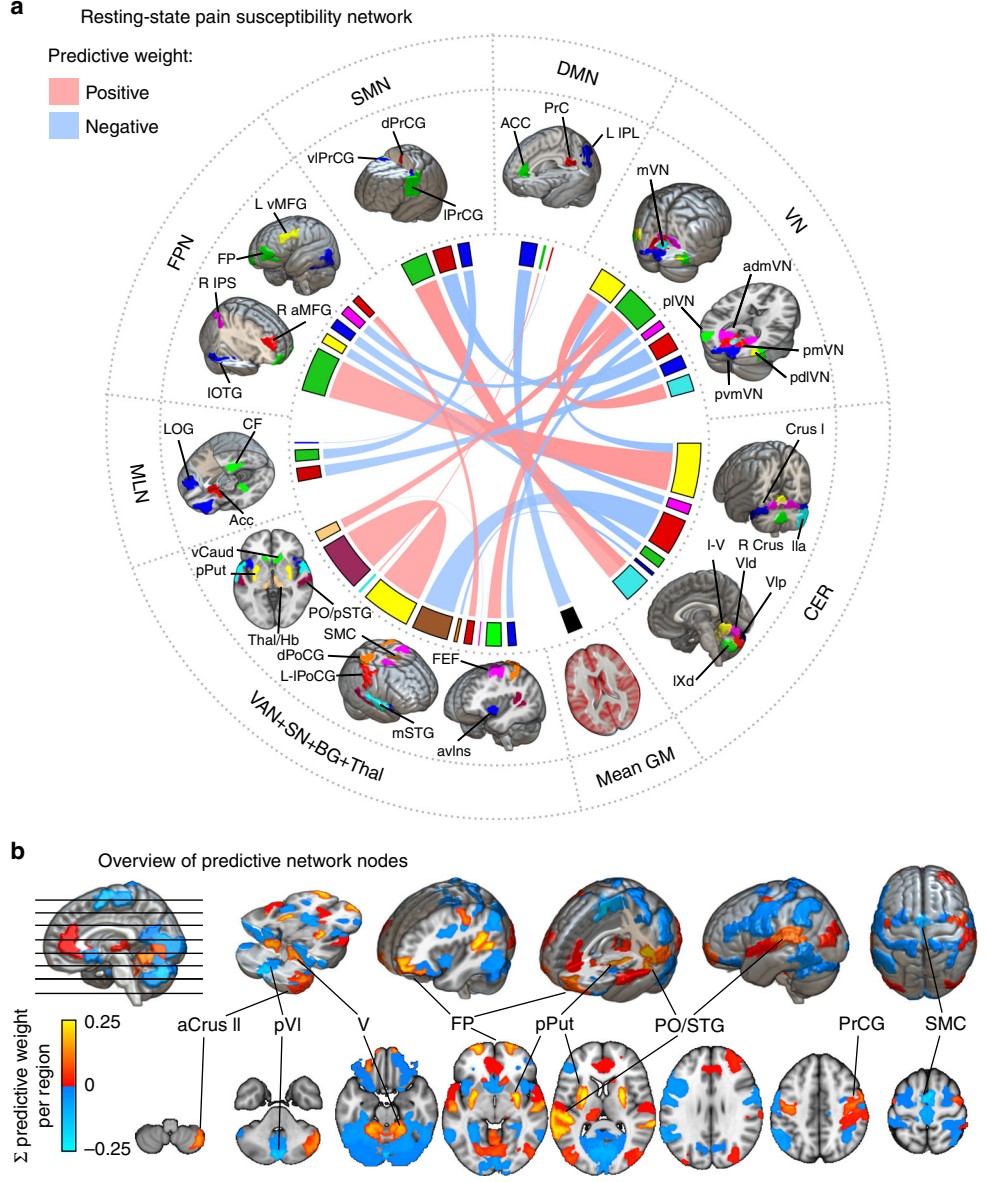

**Fig. 3 The resting-state pain sensitivity network signature. a** The predictive network of the RPN-signature. Widths of ribbons are proportional to the predictive weights of the functional connections. Network-nodes are color-coded and displayed in 3D-views. Note that, the utilised brain atlas is based on an entirely data-driven functional parcellation and is, therefore, not fully bilateral. Where laterality (L: left, R: right) is not explicitly specified, the atlas did not distinguish the region from its contralateral homolog. **b** Regional predictive strength map of the RPN-signature. Colour-bar depicts region-wise predictive strength (sum of the weights of all connections for the region, multiplied by the study-specific regional probability map). Regions with an absolute predictive strength greater than 0.1 are annotated.

prediction target. As expected from these results, the RPN-score was found to be relatively robust to the choice of pain threshold measures to construct the composite score and was significantly associated to the single thresholds. Finally, the RPN-score does not introduce bias toward/against any of involved sensory modalities and it is robust to the quality of the regional timeseries (Supplementary Note 2). These properties render the RPN-signature as a promising predictive tool for the concise, non-invasive characterization of an individual's sensitivity to pain.

The identified predictive connectivity network is relatively sparse (a predictive node has on average 1.2 links), which might be a consequence of the L1-regularisation used in the applied machine-learning pipeline. Therefore, our approach likely only captured the tip of the iceberg and the reported predictive signature should be considered as a sparse representation of the underlying true connectivity patterns. Due to the inherent variability in the feature-selection procedure other equivalent sparse-signatures might exist. However, our external validation procedure confirmed the predictive validity of the reported pattern and allows for the interpretation of the single connections.

The key nodes of the RPN-signature, such as the PO/SII, pPut, SI, dlPFC, habenula (Hb), pgACC and aIns (Fig. 3), have commonly been associated with pain[6] and corresponding networks were found to be the most predictive to individual pain thresholds[24]. However, other brain areas often associated with nociception and pain, such as the posterior insula, are not directly represented in the predictive pattern, which might be a consequence of sparse modelling. Moreover, other regions not traditionally associated with pain processing contribute to the RPN-signature.

According to current concepts, the multifaceted experience of pain results from the integration of nociception and the cognitive-emotional state of the individual[39]. Within such framework, our predictive connectivity pattern might reflect the interaction of pain-related regions with brain properties that determine the personality and cognitive and emotional and memory experiences of a given subject.

For instance, the sub-regions of the PO[40] (and the dpIns[41]) have been discussed to be specific to nociception and pain-related percepts and the involvement of the pPut in pain-related affective sensorimotor processes is well known[42]. Therefore, our observation that stronger resting-state co-activation of the pPut and SII is associated with higher sensitivity to pain might imply that the sensory aspects of salient (and possibly nociceptive) inputs have an elevated weight in this integrative process in individuals with high pain sensitivity.

Similarly, the predictive power of the resting-state connectivity of prefrontal areas to the SI and sensorimotor parts of the cerebellum (lobules I–V) with pain sensitivity may reflect the known role of the prefrontal cortex in integrating cognitive-emotional states into pain perception[23,43]. In line with this view, neuromodulation of the dlPFC was found to introduce decreased pain-related activity in sensory-motor areas[43] and the resting-state synchronisation of the prefrontal cortex and the somatosensory cortex has been reported to predict changes in pain sensitivity[26] and individual differences in pain thresholds[24].

The predictive capacity of the resting-state synchronisation between the cerebellum and sensory-motor cortical areas (PrCG, PoCG, SMC) is also remarkable and highlights the cerebellum as a promising novel target for related research.

Several connections, exerting small-to-moderate influence on the prediction, involve the occipital lobe. Some of these predictive connections could be interpreted as secondary to pain perception (analogously to the well-known deactivations of visual areas during noxious stimulation) or, alternatively, might be related to the somewhat underreported effect of visual context on pain experience[44].

In sum, our findings are in line with the notion[45] that there is no brain area that is selectively and exclusively associated with pain sensitivity and the individual variability in pain sensitivity is most probably emerges from the connectivity of multiple brain areas, which integrate an individual's sensory, cognitive and emotional state and thereby determine the overall sensitivity to pain.

While only a few papers have previously highlighted the relationship between resting-state activity and acute pain[22–26,46,47], several studies have focussed on pain anticipation[12,15–17] i.e. on a pain-free state directly preceding a painful stimulus. In summary, results from these studies suggest that the functional state and connectivity profile of the anterior insula, periaqueductal grey, anterior cingulum, cerebellum and areas of the frontoparietal network appear to reflect the individuals' momentary sensitivity to potentially painful stimuli. However, from these studies it is unclear to what degree pain sensitivity is modulated by trait-like (anxiety, pain catastrophising) or by state-like (preceding emotional appraisal, attentional or pain-specific mental states) characteristics[48]. In contrast to the short periods used in anticipatory studies, our study is based on a ten-minute-long resting-state period and predicts pain sensitivity measured several days later. Thus, our results strongly support the presence of a trait-like neural signature of pain sensitivity. However, the observed association with time between measurements and menstrual cycle also provides evidence for temporal dynamics.

While the validation and testing of the proposed predictive signature is highly reliable in terms of generalizability, here we note that the applied brain atlas, while providing full-brain coverage and a generalizable functional parcellation, still introduces a-priori constrains in laterality and precise localisation of brain regions (e.g. it does not contain the PAG and other brainstem areas). Moreover, even though the RPN-signature is not correlated with non-painful somatosensory detection thresholds, it should be noted that we did not investigate other sensory modalities such as vision or hearing.

The identified predictive network signature has important implications from a basic research and clinical perspective and paves the way for future translational research. Investigating how the resting-state pain sensitivity network is embedded into the general resting-state brain activity could extend our knowledge about the complex functional architecture of the resting brain and foster our understanding of the mechanisms by which the subjective experience of pain emerges from neural function.

While heightened pain sensitivity is a characteristic for many pain conditions[3–5], patterns of brain activity and connectivity are fundamentally distinct in experimentally evoked, acute and chronic pain[49]. A future, iterative research approach involving clinical populations promises to further improve the predictive capabilities and generalisability of the RPN-signature and may allow for assessing pain sensitivity even if reliable behavioural pain reports cannot be obtained.

In sum, the RPN-signature identified here has the potential to become a novel, non-invasive neuromarker for the supraspinal neural contribution to pain sensitivity, which is of interest in clinical pain states and especially in translational research and development of analgesic treatment strategies, where uncoupling peripheral and central mechanisms is often of crucial interest. Moreover, the RPN-signature might serve as a novel, resting network-based building block in a future pain biomarker composite signature[27].

## Methods

**General considerations.** The study design was established with careful consideration of recent recommendations, requirements and standards for neuroimaging biomarkers[50] (neuromarkers) and motivated by the following thoughts.

**Maximise predictive performance.** We employed a standardized preprocessing pipeline to ensure optimal sensitivity of the neuromarker, as sufficient effect size is a basic requirement of any clinical utility[50]. We used high-precision image alignment, incorporating individual anatomy when extracting fMRI timeseries data. Moreover, we adopted recent recommendations and protocols[51] regarding artefact reduction and optimised our workflow to meet the special needs of connectome-based analysis. We used our in-house developed, open-source python software library Pipelines Utilising a Modular Inventory (PUMI, https://github.com/spisakt/PUMI), which is based on nipype[52], a community-based Python project providing a uniform interface to existing neuroimaging software and, in part, re-used code from the C-PAC[53] and the niworkflows[54] open-source projects. A predictive modelling (machine learning) approach was utilised to exploit the rich data provided by resting-state functional brain networks and, potentially, take advantage of fMRI hyperacuity[55].

**Assessing predictive power under realistic conditions.** We used a pre-registered, external validation strategy, that strictly separated model training and performance assessment. For model training, we exclusively used data from Study 1. We conducted two independent sub-studies (Studies 2 and 3) in different research centres, with different equipment and different research staff for validation. We used a liberal alignment of research settings, allowing for a reasonable heterogeneity in procedures, equipment, imaging sequences, language of participant-researcher communication across study-centres, introducing a reasonable heterogeneity in the validation procedure to ensure generalizability.

**Ensure that prediction is driven by neural signal and is specific to pain sensitivity.** To ensure that the proposed marker of pain sensitivity is indeed driven by neural signals associated with pain sensitivity, we evaluated the correlation of the predicted score with various pre-defined (and pre-registered) confounder and validator variables.

**Ensure accessibility of results.** We applied a comprehensive pre-registration and made the source code of the method open-source and freely available for the

**Table 3 Inclusion and exclusion criteria during the recruitment process.**

| Inclusion criteria | Exclusion criteria |
|---|---|
| No chronic disease | Acute or chronic neurological endocrine, or psychiatric conditions |
| Age between 18 and 40 (target: 25) | Acute infections |
| Right-handedness | Contraindication for MRI measurement |
| Non-smoking | Regular medication intake (except contraceptive) |
| Equal gender distribution targeted | Recent use of psychotropic or analgesic substances |
| | Participation in any medication-associated study in the last 6 months |
| | Wounds, scars or any other skin conditions (e.g. neurodermitis) which could affect QST measurements on the forearm and the hands |

**Table 4 MRI scanner and sequence parameters for each centre.**

| | Study 1 | Study 2 | Study 3 |
|---|---|---|---|
| **General** | | | |
| Scanner | Philips Achieva X 3 T | Siemens Magnetom Skyra 3 T | GE Discovery MR750w 3 T |
| Head coil | 32-channel | 32-channel | 20-channel |
| **Anatomical scan** | | | |
| Weighting | T1 | T1 | T1 |
| Sequence | MPRAGE | MPRAGE | 3D IR-FSPGR |
| TR | 8500 ms | 2300 ms | 5.3 ms |
| TE | 3.9 ms | 2.07 ms | 2.1 ms |
| Resolution | $1 \times 1 \times 1 mm^3$ | $1 \times 1 \times 1 mm^3$ | $1 \times 1 \times 1 mm^3$ |
| FOV | $256 \times 256 \times 220 mm^3$ | $256 \times 256 \times 192 mm^3$ | $256 \times 256 \times 172$ |
| **Resting state fMRI** | | | |
| Weighting | T2* | T2* | T2* |
| Sequence | GE EPI | GE EPI | GE EPI |
| TR | 2500 ms | 2520 ms | 2500 ms |
| TE | 35 ms | 35 ms | 27 ms |
| Flip angle | 90 | 90 | 81 |
| Phase ENC. DIR | COL | A>>P | A>>P |
| FOV | $240 \times 240 \times 132$ | $230 \times 230 \times 132$ | $96 \times 96 \times 44$ |
| Num. of slices | 40 | 38 | 44 |
| Slice thickness | 3 mm | 3 mm | 3 mm |
| GAP | 0.3 mm | 0.48 mm | 0 mm |
| Slice order | Interleaved | Interleaved | Interleaved |
| In-plane res. | $3 \times 3mm^2$ | $2.45 \times 2.45mm^2$ | $3 \times 3mm^2$ |
| Acceleration | SENSE 3× | GRAPPA 2× | ASSET 2× |
| Fat suppress | SPIR | Fat.sat. | Fat. Sat |
| Num. of vols | 200 | 290 | 240 |
| Dummy Scans | 5 | 5 | 0 |
| Scanning time | 8 min 37 sec | 12 min 11 sec | 10 min |

community. Moreover, we provide a platform-independent, easy to use docker container, which provides the opportunity to use our predictive model as a research product[50], to obtain out-of-the box pain sensitivity predictions form any appropriate imaging datasets.

**Participants.** A total of $N = 116$ healthy, young volunteers were involved in three sub-studies. Age and sex of the participants is reported in Supplementary Table 1. Study 1 involved $N_1 = 39$ participants (the same sample as in ref. [8]). It was performed at the Ruhr University Bochum (Germany) by MZ and TSW and used as the training sample for the machine-learning-based prediction of pain sensitivity and additionally, served as a basis for the internal validation of the prediction. Studies 2 and 3 ($N_2 = 48$, $N_3 = 29$) were performed at the University Hospital Essen (Germany) by FS and TS and at the University of Szeged (Hungary) by BK and TK, respectively, and served as samples for external validation. Inclusion criteria and exclusion criteria were largely identical in all three centres and are listed in Table 3. Recruitment and reimbursement policies varied across centres; participants received 20 €/h in Studies 1 and 2 and no reimbursement in Study 3.

Metal implant, unremovable piercing, peacemaker, tattoo in head/neck position, pregnancy or known claustrophobia were considered as contraindication for MR measurement. Participants were required to abstain from consuming caffeine two hours before experiments (except in Study 3) and from consuming alcohol on the day of testing and the previous day.

The study was conducted in accordance with the Declaration of Helsinki, complies with all relevant ethical regulations for work with human participants and was approved by the local or national ethics committees (Register Numbers: 4974-14, 18-8020-BO and 057617/2015/OTIG at the Ruhr University Bochum,

University Hospital Essen and ETT TUKEB Hungary, respectively.) All participants gave written informed consent before testing.

Imaging and quantitative sensory testing (QST) were performed on the same day in Study 1 and in average 2–3 days apart in Studies 2 and 3 (see Supplementary Table 1 for details). MRI measurement always preceded the QST session.

**Measures—functional MRI.** High-resolution anatomical and open-eyed resting-state fMRI measurements were acquired from all participants. Scanning parameters (including equipment) varied across centres and are listed in Table 4. During measurements, participants were instructed to lie still and relaxed, without falling asleep, and avoid any movement. Foam padding, and in Studies 1 and 2, pneumatic pillows were used to restrict head movements. All anatomical MRI measurements were screened for incidental findings.

**Measures—QST.** Heat (HPT), cold (CPT) and mechanical (MPT) pain thresholds were acquired according to the QST protocol[28]. Warmth (WDT), cold (CDT) and in Study 2 and Study 3, mechanical (MDT) detection thresholds were obtained as additional control measures. All sensory measurements were obtained from the palmar left forearm, proximal to the wrist crest. Within the QST framework, thermal thresholds are determined using a method of limits. To this end, increasing and decreasing temperatures were applied to the skin with an MSA thermal stimulator (Somedic, Hörby, Sweden) in Study 1 and Pathway thermal stimulators (Medoc Ltd., Ramat Yishai, Israel) in Studies 2 and 3. In all studies, ATS thermodes were used on a skin surface of $30 \times 30mm$, with a baseline temperature of 32 °C. Participants were instructed to indicate the onset of pain by button press. For all thermal thresholds 6, rather than 3 (as in the original protocol)[28], stimulus repetitions were performed to reduce between-subject variance. Furthermore, the first measurement was discarded from analysis as a test stimulus. HPT and CPT were calculated as the arithmetic means of the five remaining threshold temperatures. MPTs and MDTs were determined using a staircase method. Five increasing and five decreasing trains of pinprick (MRC Systems, Heidelberg, Germany) stimuli were applied to the palmar left forearm in an alternating fashion, whereas the participant was instructed to categorize the stimuli as noxious, or non-noxious. Mechanical detection threshold was assessed analogously with von Frey filament stimulations. MPT and MDT were computed as the log-transformed geometric mean force determined in five ascending and descending staircase-thresholding-runs.

**Additional measures.** Age, sex, self-reported height, weight and, for female participants, the date of the first day of the last menses and the use of contraceptives, was recorded prior to all measurements. Additionally, self-reported weekly alcohol consumption and level of education (primary school, secondary school, university) was recorder for Studies 1 and 2. Before the QST, participants filled out the Pain Sensitivity Questionnaire (PSQ)[56], the Pain Catastrophizing Scale (PCS)[57], the State-Trait Anxiety Inventory (STAI)[58], the short German version of the Depression Scale (ADS-K, Center for Epidemiologic Studies)[59] and, additionally in Studies 2 and 3, the Pittsburgh Sleep Quality Index (PSQI)[60] and the perceived stress questionnaire (PSQ20)[61]. In Studies 2 and 3, blood pressure was measured both before the MRI and the QST measurements. Moreover, for Sample 1, $T_{50}$ values were available from a parallel experiment performed on the day before fMRI testing. $T_{50}$ represents the temperature (in °C) necessary to induce a heat-pain rating of 50 (on a scale ranging from 0, no pain to 100 unbearable pain). $T_{50}$ values were obtained from a non-linear (second-order polynomial) interpolation of ratings obtained in response to 15 tonic heat-pain stimuli (duration: 16 s) between 42.5 °C and 48 °C, presented in pseudo-randomized grid-search fashion.

**Calculation of pain sensitivity.** The target variable for the prediction was a single composite measure of individual pain sensitivity summarizing HPT, CPT and MPT as defined in ref. [8].

In Study 1, HPT, CPT and MPT were Z-transformed (mean centred and standardized) and HPT, as well as MPT were inverted (multiplied by $-1$), so that higher Z-values denoted higher pain sensitivity. Then, the arithmetic mean of the Z-transformed variables was computed for each participant and defined as pain-sensitivity score. In Studies 2 and 3, the same procedure was applied, except that Z-transformation was based on the population-mean and standard deviation of Study

1, to ensure that the same scale was used across studies. Extreme QST values were defined using the normative 95% percentiles reported in ref. [28]; participants showing extreme HPT, CPT or MPT values in at least two of the three modalities were excluded. This screening resulted in excluding 0, 3 and 2 participants in Samples 1, 2 and 3, respectively (Supplementary Table 2).

**fMRI preprocessing**. As fMRI-based functional connectivity is susceptible to in-scanner motion artefacts[62,63], appropriate preprocessing and signal cleaning is key to successful connectivity-based prediction. Resting-state functional MRI data were preprocessed identically in all three studies. The applied, nipype-based workflow is depicted on Supplementary Fig. 1. It utilised third-party neuroimaging software, code adapted from the software tools C-PAC[53] and niworkflows[54], and in-house python routines.

Brain extraction from both the anatomical and the structural images, as well, as tissue-segmentation from the anatomical images was performed with FSL bet and fast[64]. Anatomical images were linearly and non-linearly co-registered to the 1mm-resolution MNI152 standard brain template brain with ANTs[65] (see https://gist.github.com/spisakt/0caa7ec4bc18d3ed736d3a4e49da7415 for source code).

Functional images were co-registered to the anatomical images with the boundary-based registration technique of FSL flirt. All resulting transformations were saved for further use. The preprocessing of functional images happened in the native image space, without resampling. Realignment-based motion-correction was performed with FSL mcflirt. The resulting six head motion estimates (3 rotations, 3 translations), their squared versions, their derivates and the squared derivates (known as the Friston-24-expansion[66]) was calculated and saved for nuisance correction. Additionally, head motion was summarised as framewise displacement (FD) timeseries, according to Power's method[63], to be used in data censoring and exclusion. After motion-correction, outliers (e.g. motion spikes) in timeseries data were attenuated using AFNI despike[67]. The union of the eroded white-matter maps and ventricle masks were transformed to the native functional space and used for extracting noise-signal for anatomical CompCor correction[68].

In a nuisance regression step, 6 CompCor parameters (the 6 first principal components of the noise-region timeseries), the Friston-24 motion parameters and the linear trend were removed from the timeseries data with a general linear model. On the residual data, temporal bandpass filtering was performed with AFNI's 3DBandpass to retain the 0.008–0.08 Hz frequency band. The prior use of AFNI's despike is expected to attenuate aliasing of residual motion artefacts into the neighbouring time-frames during bandpass filtering[69]. To further attenuate the impact of motion artefacts, potentially motion-contaminated time-frames, defined by a conservative FD > 0.15 mm threshold, were dropped from the data (known as scrubbing the data)[70]. Participants were excluded from further analysis if the mean FD exceeded 0.15 mm, or when more then 30% of frames were scrubbed. This resulted in exclusion of 4, 8 and 7 participants in Samples 1, 2 and 3, respectively (Supplementary Table 2). Quality-control (registration-check, carpet-plots, see e.g. Supplementary Figs. 2–4) was performed throughout the workflow.

**Functional connectivity analysis**. The 122-parcel version of the MIST[71] multi-resolution functional brain atlas and grey-matter masks obtained from the anatomical image were transformed to the native functional space. This atlas (constructed with by the BASC method, i.e. bootstrap analysis of stable clusters) was recently shown to perform well in connectivity-based predictive modelling[72]. Native-space atlas regions were masked with the grey-matter masks that were obtained from the anatomical image and transformed to functional space previously. With this atlas-individualization technique, the final regional signal will originate—with a high probability—from grey-matter voxels for each subject (which we carefully checked manually for all subjects), while with the conventional method, a variable ratio of grey- and white-matter voxels are included for every subject. Therefore, inputting information from the tissue-segmentation process is expected to decrease subject-to-subject variability (see Supplementary Fig. 5 for examples). Voxel-timeseries were averaged over these individualised MIST regions and, together with the mean grey-matter signal, retained for graph-based connectivity analysis.

Regional timeseries were ordered into large-scale functional modules (defined by the 7-parcel MIST atlas) for visualization purposes (Fig. 1). Partial correlation was computed across all pairs of regions (and global grey matter), as implemented in the nilearn[73] python module. Partial, rather than simple correlations were used to rule out indirect connectivity[74]. Our graph-modelling approach ensured that the global grey-matter signal is handled as a confound during computing the partial correlation coefficients but, at the same time, also considered it as a signal of interest, as it may represent vigilance related processes[75]. Partial correlation coefficients were organised to 123 by 123 (122 regions + global grey-matter signal) symmetric connectivity matrices. The upper triangle of these matrices was used as the feature space for machine-learning-based predictive modelling.

**Predictive model training and validation**. Whole-brain resting-state functional connectivity data of study 1 ($N_1 = 35$, after all exclusions, as in ref. [8], Supplementary Table 2) was used as the input feature-space ($P = 7503$ features per participant) to predict individual pain sensitivity scores, leading to a large P—small N setting.

We constructed a machine-learning pipeline (https://github.com/spisakt/RPN-signature/blob/master/PAINTeR/model.py) in scikit-learn[76], consisting of robust feature scaling (removes the median and scales with data quantiles), pre-selection of features[77], selecting the K best features with strongest relationships to the target variable and an Elastic Net regression model[78] (a linear model with combined $L_1$ and $L_2$-norms as regulariser). The use of elastic net was a decision drawn prior to the analysis. Our main motivation to choose elastic net was that it allows to optimize sparsity (L1 vs. L2 regularization) as a hyperparameter, so that we did not have to make any a-priory assumptions about the sparsity of the discriminative ground truth (see ref. [79] for rationale). To summarise, free hyperparameters of the machine-learning pipeline were the number of pre-selected features (K), the ratio of the $L_1/L_2$-regularization and the weight (alpha) of regularisation. Hyperparameters were optimised with a grid-search procedure and negative mean squared error as cost function. Values for K ranged from 10 to 200 with increments of 5, and included [0.1, 0.5, 0.7, 0.9, 0.95, 0.99, 0.999] for the $L_1/L_2$ ratio [0.001, 0.005, 0.01, 0.05, 0.1, 0.5] for alpha. Hyperparameter optimisation was performed in a leave-one-participant-out cross-validation (internal validation phase). Cross-validation incorporated the complete machine-learning pipeline to avoid introducing dependencies between the training and test samples. Note that fMRI preprocessing was independent between subjects, thus not included in the cross-validation. Optimal hyperparameters were found to be K = 25, $L_1/L_2$-ratio = 0.999 and alpha = 0.005.

External validation was performed by applying the RPN-signature on the fMRI data of Studies 2 and 3 ($N_2 = 37$, $N_3 = 19$, after exclusions, Supplementary Table 2), simply by applying the feature transformation (scaling) obtained on Sample 1 and then calculating the dot product between individual connectivity matrices and the non-zero feature weights obtained in Sample 1. The resulting predictions were compared with the observed QST-based pain sensitivity scores by calculating mean absolute error (MAE), mean squared error (MSE) and explained variance. Permutation-based p-values were obtained for all three measures, using the mlxtend python package. Moreover, bootstrapping with conditional coverage[80] was used to provide p-values for predictive connectivity weights to aid interpretation. We constructed 10000 bootstrap samples (with replacement), with a size equal to the original sample, consisting of paired brain and outcome data. The predictive model with the optimal hyperparameters was fitted to each sample. Uncorrected P-values were calculated for each selected connection based on the proportion of weights below or above zero, as in e.g. ref. [30]. Note that the interpretation of these p-values and confidence intervals (Supplementary Table 4) remains limited as they are conditioned to the feature-selection procedure.

**Confounder analysis**. To explore potential confounding variables, the predicted pain sensitivity-scores (or cross-validated predictions in case of Sample 1) were contrasted to mean and median FD, the percentage of scrubbed volumes, systolic and diastolic blood pressure before both the MRI and QST measurement (as blood pressure was earlier reported[81] to be associated with sensitivity to mechanical pain), the time delay between MRI and QST testing (to test for temporal stability of the prediction), age, sex, BMI, number of days since the first day of the last menses, alcohol consumption (units/week), level of education, state and trait anxiety (STAI), score of depressive symptoms (ADS-K), self-reported pain sensitivity (PSQ) and pain catastrophising (PCS), perceived stress (PSQ20), quality of sleep (PSQI), and non-noxious QST detection thresholds (CDT, WDT and MDT, where available). Moreover, in Study 1 predictions were compared to T50-values and MR spectroscopy-based GABA and Glutamate/Glutamine levels in pain-processing brain regions (see ref. [8] for details). Associations were tested with permutation-based linear models.

**Visualization of the predictive network**. The predictive interregional connections highlighted by the non-zero regression coefficients of the RPN-signature were displayed as a ribbon plot using the R-package circlize (Fig. 3). Corresponding individualised brain region masks were transformed back to standard space to create a study-specific regional probability map (reflecting co-registration accuracy and individual variability in morphology). Probability maps were multiplied by the sum of corresponding regression coefficients to create a regional predictive strength map, which was then visualised with FSLeyes and MRIcroGL.

(https://www.mccauslandcenter.sc.edu/mricrogl) (Fig. 3). The analysis of large-scale resting-state network-involvement (as defined by the MIST[71] brain atlas) was performed by summarising and Z-transforming the voxel values across the seven regions-of-interest. Polar plot was made with the R-package ggplot2.

**Software availability**. The RPN-signature scores can be computed based on structural and resting-state functional datasets by the software tool with the same name. The RPN-signature software tool consists of the described MRI processing pipeline and the functional connectome-based predictive model. It is available as source code at https://github.com/spisakt/RPN-signature. As the software follows the Brain Imaging Data Structure (BIDS)[82] and the BIDS-App specification, it provides a standard command line interface and relies on Docker-technology. The docker image is deposited on Docker Hub: (https://cloud.docker.com/repository/docker/tspisak/rpn-signature) and does not depend on any software outside the container image. This, together with the fully transparent continuous integration-based development and automated tagging and versioning, enhances software availability and supports reproducibility of RPN-signature results.

**Reporting summary**. Further information on research design is available in the Nature Research Reporting Summary linked to this article.

## Data availability

Processed data (regional timeseries) and source code are deposited at https://github.com/spisakt/RPN-signature. The source data underlying Fig. 2 are provided at the same website and as a Source Data file. Raw imaging data is available at openneuro.org (ds001900).

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

## Acknowledgements

We are thankful to Katja Wiech (University of Oxford, UK), Tor Wager (Presidential Cluster in Neuroscience and Department of Psychological and Brain Sciences, Dartmouth College, USA) and Markus Ploner (Technical University of Munich, Germany) for their valuable insights and comments on the manuscript. This research was supported by the Mercator Research Center Ruhr (MERCUR) and by the Deutsche Forschungsgemeinschaft (DFG, German Research Foundation) – Projektnummer 316803389 – SFB 1280. B.K. was supported by the UNKP-18-3 New National Excellence Program of the Ministry of Human Capacities, Hungary. T.S.W. is supported by the DFG (SFB874-A8).

## Author contributions

T.S., M.Z. and U.B conceived the study, Study 1 was and performed by M.Z. and T.S-W. Studies 2 and 3 were performed by F.S. and T.S. and by B.K. and Z.T.K., respectively. T.S. and B.K. conducted the analyses, T.S., U.B. wrote the manuscript. M.Z., Z.T.K, B.K. contributed to the interpretation, as well as manuscript revision. The whole study was supervised by T.S. and U.B.

## Competing interests

The authors declare no competing interests.
