## [Peer Review File · Nature Communications]

Reviewers' comments:

Reviewer #1 (Remarks to the Author):

Spisak et al. report a pattern of resting fMRI correlated inter-regional activity that is associated with individual differences in pain sensitivity assessed with QST in 3 independent cohorts. Accounting for various potential confounding factors, they apply rigorous methods and external validation to demonstrate generalizability of the observed activity pattern.

Understanding the variability in pain perception/sensitivity and individual brain organization is an important goal, and the authors make progress toward this end in a unique and potentially generalizable manner. The critical external validation analyses particularly set this study apart from others in the field demonstrating associations between resting fMRI and pain-related measures. However, I have concerns regarding the strength of the evidence presented as well as the authors' interpretations, given the following issues:

Major points:

1. In the main analyses, pain sensitivity was considered as a composite score based on heat, cold and mechanical pain thresholds. However, it could be that these psychophysical measures partially diverge from one another and that unique brain patterns could be associated with each. Were the pain thresholds across categories correlated with one another within each of the 3 cohorts? The authors report a breakdown of the three categories for studies 2 and 3. However, those results do not appear to be consistent (Figure 2D). For example, in Study 2, a negative correlation is reported for HPT that is not seen in Study 3. Moreover, in the main text, the r value is reported as positive (0.48), whereas in the figure the same relationship is shown as negative (-0.48). In the figure, it is also unclear why the authors display the correlations with quartile-based subgroups rather than showing all data points as done for the composite scores. Overall, I feel that more clarity and transparency is needed regarding the convergence/divergence of the distinct pain threshold measures and their relationships with functional connectivity patterns.

2. While pain sensitivity is an important index of individual differences in pain perception, there are several additional psychophysical measures also relevant to pain coping that are not considered here (e.g. pain tolerance, temporal summation of pain). It is thus unclear whether a neural pattern reflecting pain sensitivity alone is a general marker of "susceptibility to pain" as implied by the authors. Unless this issue can be addressed, feel that the authors should tone down their suggested implications.

3. In defining features, the authors considered only time-averaged (or "static") functional connectivity (FC) across whole fMRI scans. However, the resting fMRI data are very rich, and there are additional features that could have utility in predicting behavior. For example, Cheng et al (2018 Pain) applied similar machine learning approaches within chronic pain patients and showed that features based on both static and dynamic (time-varying) FC contributed unique variance to measures of self-reported pain. Thus, given increasing evidence for the relevance of resting fMRI measures beyond static FC to behavior (including pain), it is possible that features missed in the authors' analyses are important in predicting pain sensitivity.

4. As presented, the analyses suggesting overlap between the "RPN signature" and the NPS and SIIPS1 patterns are not convincing. As shown in Figure 4, it is clear that many RPN areas do not overlap with these other patterns (and vice versa), but the authors mainly focus only on the potential similarities. Given that there is no null model for this analysis, the authors have not ruled out that some or all of the overlap could be due to chance. Moreover, given the distinctness of the data and analyses underlying the compared maps (as acknowledged by the authors), it is not clear how this comparison should be interpreted.

5. In several instances (including the abstract), the authors imply clinical relevance of their findings and suggest that the RPN pattern may be applied to chronic pain patients. However, as the data presented are based on healthy subjects, and given the various factors that could affect generalizability to patients, I strongly suggest that the authors' tone down claims about clinical relevance.

Minor points:

- I feel that the manuscript is repetitive in many instances, especially when describing general methodological issues in the field rather than the specific goals of the study. I suggest that the authors remove repetition and tighten the text about these general methodological issues (e.g. about reproducibility and validation). For example, there is no need to mention within the main Results text that the pipeline is BIDS-compatible and to include repeated references to the same hyperlinks in different sections.

- The terminology "direct functional connectivity," used in several instances, is ambiguous. I understand that this is used because of the partial correlation analyses, but still the term "direct" may imply directionality (which cannot be addressed even based on the partial correlations), and I therefore suggest that this be removed or changed/better described.

- I suggest mentioning earlier on in the Results that pain sensitivity was based on a composite measure of heat, cold, and mechanical thresholds (it took me a while to realize this upon first read).

- The reported QST-based sensitivity values in arbitrary units are difficult to interpret. Could the authors be more informative about these values, compared to those reported in normative datasets, within the main text?

- In discussion, the authors state "...only a few papers have previously highlighted the relationship between resting-state activity and acute pain." There are several more references in this area not cited by the authors: e.g. Riedl et al 2011 Neuroimage; Cheng et al 2015 J Neurosci & 2017 Neuroimage

- In discussion, the authors suggest that their findings "corroborate" that subregions of the PO and dpIns are specific to nociception/pain. The reported resting FC analyses do not directly corroborate this, and thus I suggest removing or changing this wording.

- The first section of the Methods ("study design considerations") could be considerably shortened, as many of the details are not necessary for reproducing the present study.

Reviewer #2 (Remarks to the Author):

This is an elegant paper. The authors generate a pain sensitivity brain resting state network and validate the results in 2 studies performed in different institutions. These results are compelling and set a new level of rigor that researchers in the field should emulate. Technically the study is excellent. There are however important weaknesses that need attention and would render the study more solid and understandable.

1) The groupings of brain regions with each other is unclear. In figure 3 why is VAN+BG+Th grouped together? Other groupings are based on RSNs, while some are purely anatomical, like the cerebellum. At least the reasoning of these parcellations needs clarity.

2) The overall functional anatomical parcellation also needs further explanation. Perhaps illustrated in the supplement.

3) It is also not clear how variable a given parcel may be from one subject to the next. As the

functional networks are based on subject space anatomy, there must be distinct extents of variations for individual parcels. Please clarify and expound.

4) Although the authors repeatedly indicate that their signature is distinct and not directly comparable to pattern-based signatures for predicting presence of pain, they have a figure where these signatures are compared to their own. This is really a comparison between apples and oranges and simply leads to confusion. I strongly urge removing the figure and avoiding the comparisons. Activity and connectivity are different concepts and cannot be lumped together.

5) A large part of the discussion again suffers from the temptation of lumping connectivity with activity. The specific brain regions and their link to pain perception are discussed although this concept is inconsistent with the network being the determining factor. The discussion of the role of S1 or S2, for example, with its specific connectivity is simply an over interpretation. The fact is that the identified network seems very distinct. For example it does not include posterior insulation, a region being debated to have some specificity for nociceptive information encoding in the cortex. In contrast, very large portions of the cerebellum seem involved in their signature and this is novel and perhaps uniquely linked with pain sensitivity.

6) A far simpler interpretation of the network is that it is a global reflection of how nociceptive information interacts with brain properties that determine the personality and cognitive and emotional and memory experiences of a given subject. Thus, searching in the literature for evidence that pieces of the network have some direct relationship with pain/nociception encoding regions of the brain remains unconvincing and perhaps points in the wrong direction.

Overall the study remains very exciting. However, the results are overinterpreted and compared to outcomes that are not relevant. The proposed changes clarifications should be easy to implement and would enhance the paper.

Reviewer #3 (Remarks to the Author):

This is a very interesting topic. My main concerns are as follows:

1. The aim of the study is to using resting state functional connectivity to predict pain sensitivity. They used three pain threshold measurements in the manuscript. It would be helpful to demonstrate the association of these three measurements and discuss the reliability of pain sensitivity.

2. The authors used a single composite measure by averaging HPT, CPT, and MPT. Because heat pain and mechanical pain activate different fibers, it is likely that brain patterns encoding HP and MP are different. To generalize the results, the authors may want to 1) compare the predictive patterns for different pain modalities and/or 2) build the model from one modality and test on other modalities.

3. The rationale of using MIST for brain parcellation is unclear. Since MIST is a very new atlas, I wonder if the authors have validated their findings based on other well-established brain parcellation strategies (e.g., Yeo atlas, Power atlas).

4. Line 786. How did the authors pre-select features? Ref 42 may not be the one to cite. If K was selected from 10 to 200, this number is \ll original feature space ($k=7503$), and if the optimal K was 25, which is smaller than N ($N_1=35$), I would question the necessity of using elastic net. A recent benchmark study by Dadi and colleagues [Neuroimage, 192 (2019) 115-134] demonstrated that non-sparse linear methods outperformed sparse linear methods in connectome-based prediction tasks. The authors need to clarify the feature selection procedure.

5. The details of external validation are missing. Did the authors take dot product between connectivity patterns and non-zero feature weights?

6. Statistical significance for MAE is not provided. The authors may consider permutation testing to assess the performance of MAE.

7. Results in Figure 2D suggest that the RPN may fail to generalize to single modalities (e.g., mechanical pain in the upper panel and heat pain in the lower panel). Those results may be largely affected by the collinearity between the composite scores and single-modality scores.

8. The importance of the connectivity in the prediction was represented by the raw coefficient

weights but not statistical values. Statistical assessments for those coefficients are preferred. Details can be found in Wager's NPS in Figure 1 NEJM 2013.

9. Line 322. It is overstated that the RPN is specific to pain sensitivity and is not driven by the general sensitivity to sensory stimuli. In this study, other sensory modalities have not been tested.

Minor:

1. The authors should check all their cited references. A lot of references do not match the information they were trying to provide. Two examples: Line 63, Ref 8. Line 289, Ref 33.

Point-by-point response to reviewers' comments

Reviewer #1

Spisak et al. report a pattern of resting fMRI correlated inter-regional activity that is associated with individual differences in pain sensitivity assessed with QST in 3 independent cohorts. Accounting for various potential confounding factors, they apply rigorous methods and external validation to demonstrate generalizability of the observed activity pattern.

Understanding the variability in pain perception/sensitivity and individual brain organization is an important goal, and the authors make progress toward this end in a unique and potentially generalizable manner. The critical external validation analyses particularly set this study apart from others in the field demonstrating associations between resting fMRI and pain-related measures.

We thank the reviewer for acknowledging the relevance of an individual's pain sensitivity and its link to brain organization and for highlighting the rigorous methodological approach used in our study. We are grateful for the critical points raised by this reviewer which were very helpful to strengthen and sharpen the main conclusions drawn from our study.

However, I have concerns regarding the strength of the evidence presented as well as the authors' interpretations, given the following issues:

Major points:

1. In the main analyses, pain sensitivity was considered as a composite score based on heat, cold and mechanical pain thresholds. However, it could be that these psychophysical measures partially diverge from one another and that unique brain patterns could be associated with each. Were the pain thresholds across categories correlated with one another within each of the 3 cohorts? The authors report a breakdown of the three categories for studies 2 and 3. However, those results do not appear to be consistent (Figure 2D). For example, in Study 2, a negative correlation is reported for HPT that is not seen in Study 3. Moreover, in the main text, the r value is reported as positive (0.48), whereas in the figure the same relationship is shown as negative (-0.48). In the figure, it is also unclear why the authors display the correlations with quartile-based subgroups rather than showing all data points as done for the composite scores. Overall, I feel that more clarity and transparency is needed regarding the convergence/divergence of the distinct pain threshold measures and their relationships with functional connectivity patterns.

We would like to thank the reviewer for this comment. The overall aim of our study was to evaluate the link between brain organization and pain sensitivity in a general, modality-independent way, and to specifically focus on the “common ground” across different pain modalities. This is why we used the composite and, to a certain degree, “sensory modality independent” estimate of pain sensitivity, as already reported in (Zunhammer et al., 2016). The former version of Figure 2 may have distracted from this overall goal. Therefore, we deleted part D from Fig.2., changed the caption and the text (line 122) accordingly and added a supplementary figure (Figure S5) with the correlations between HPT, CPT and MPT in all cohorts. We also added (lines, 43 and 63) a clarification of our motivations into the Introduction and the start of the Results.

The observation that the correlation between HPT and the RPN-score differs in Study 2 and 3 is correct, however please note that the error margin for this correlation in Study 3 (which had the lowest N) was so wide that no conclusions should be drawn (e.g. a negative correlation is easily within the error interval off).

We also fixed the typo in the text regarding the sign of the r-value between HPT and the RPN-score in Study 2.

Importantly, our decision to search for a composite, largely modality-independent signature has the advantage of mitigating modality-specific confounds (like reaction times which confound the thermal thresholds but not the mechanical thresholds). Nevertheless, we certainly agree, that training and comparing models for HPT, CPT and MPT would substantially aid our understanding of how neural processing differs in pain modalities and how the general, modality-independent trait of pain sensitivity is established. Given that such an analysis is expected to provide several new insights and it would, therefore, require an extensive discussion, we feel that it is out of the scope of the present study and provides an excellent topic for further research.

2. While pain sensitivity is an important index of individual differences in pain perception, there are several additional psychophysical measures also relevant to pain coping that are not considered here (e.g. pain tolerance, temporal summation of pain). It is thus unclear whether a neural pattern reflecting pain sensitivity alone is general marker of “susceptibility to pain” as implied by the authors. Unless this issue can be addressed, feel that the authors should tone down their suggested implications.

The reviewer correctly highlights that we aimed specifically at targeting individual differences in pain sensitivity, instead of related but distinct phenomena like coping, tolerance, summation, etc. Throughout the revised version of the manuscript we have now toned down any statements about “susceptibility to pain” which we have used, indeed somewhat misleadingly, as a synonym to “sensitivity to pain”.

3. In defining features, the authors considered only time-averaged (or “static”) functional connectivity (FC) across whole fMRI scans. However, the resting fMRI data are very rich, and there are additional features that could have utility in predicting behavior. For example, Cheng et al (2018 Pain) applied similar machine learning approaches within chronic pain patients and showed that features based on both static and dynamic (time-varying) FC contributed unique variance to measures of self-reported pain. Thus, given increasing evidence for the relevance of resting fMRI measures beyond static FC to behavior (including pain), it is possible that features missed in the authors’ analyses are important in predicting pain sensitivity.

We fully agree that the prediction of pain sensitivity might significantly benefit from including features of dynamic connectivity. However, to be in line with our pre-registration, we trained our predictive model exclusively on the sample of Study 1, i.e., we had to deal with a small N – large P setting. For this reason, we intentionally avoided any type of expansion of the feature set in the current study. We plan to incorporate these dynamic connectivity components in future larger-scale studies.

4. As presented, the analyses suggesting overlap between the “RPN signature” and the NPS and SIIPSI patterns are not convincing. As shown in Figure 4, it is clear that many RPN areas do not overlap with these other patterns (and vice versa), but the authors mainly focus only on the potential similarities. Given that there is no null model for this analysis, the authors have not ruled out that some or all of the overlap could be due to chance. Moreover, given the distinctness of the data and analyses underlying the compared maps (as acknowledged by the authors), it is not clear how this comparison should be interpreted.

We agree that the conceptual difference between activity and connectivity and the lack of a null model renders this comparison difficult. We have completely removed Figure 4 from the manuscript and deleted the related sentences from the text.

5. In several instances (including the abstract), the authors imply clinical relevance of their findings and suggest that the RPN pattern may be applied to chronic pain patients. However, as the data presented are based on healthy subjects, and given the various factors that could affect generalizability to patients, I strongly suggest that the authors’ tone down claims about clinical relevance.

We agree that the discussion of the potential clinical relevance of our findings was at several places too bold. We are fully aware that there are fundamentally distinct patterns of brain activity in experimentally evoked, acute and chronic pain and that future research has to test how the pattern identified in our study may apply or be developed towards a measure related to clinical conditions including chronic pain. We have consequently tuned down any relation to clinical pain states and emphasized the need for further research towards this direction in the outlook section.

Minor points:

- I feel that the manuscript is repetitive in many instances, especially when describing general methodological issues in the field rather than the specific goals of the study. I suggest that the authors remove repetition and tighten the text about these general methodological issues (e.g. about reproducibility and validation). For example, there is no need to mention within the main Results text that the pipeline is BIDS-compatible and to include repeated references to the same hyperlinks in different sections.

We carefully removed redundant thoughts throughout the manuscript (including the hyperlinks) and significantly shortened the description of general methodological considerations.

- The terminology “direct functional connectivity,” used in several instances, is ambiguous. I understand that this is used because of the partial correlation analyses, but still the term “direct” may imply directionality (which cannot be addressed even based on the partial correlations), and I therefore suggest that this be removed or changed/better described.

We removed all occurrences of the word “direct”, when referring to connectivity.

- I suggest mentioning earlier on in the Results that pain sensitivity was based on a composite measure of heat, cold, and mechanical thresholds (it took me a while to realize this upon first read).

Thank you for this suggestion. In the revised text we mention this important information earlier (line 64), which indeed improves the readability of the manuscript.

- The reported QST-based sensitivity values in arbitrary units are difficult to interpret. Could the authors be more informative about these values, compared to those reported in normative datasets, within the main text?

As the scales of the components (HPT, CPT, MPT) of the composite scores are very different (measured in units of °C and mN and spreading to scales differing with almost an order of magnitude) and normalizing them before making the composite score is an elemental step. Unfortunately, that makes it hard to interpret the values of the composite score. Nevertheless, we can plot how individual HPT, CPT and MPT values are typically mapped to the composite score. We added such a figure into the supplementary material (Supplementary Figure S7). We linked this figure in the main text and noted that it can aid the interpretation of the values of the composite pain sensitivity score.

- In discussion, the authors state "...only a few papers have previously highlighted the relationship between resting-state activity and acute pain." There are several more references in this area not cited by the authors: e.g. Riedl et al 2011 Neuroimage; Cheng et al 2015 J Neurosci & 2017 Neuroimage

We are thankful to the reviewer for these very relevant references which are now integrated into the manuscript (e.g. in the Introduction at line 25, and at many places in the Discussion).

- In discussion, the authors suggest that their findings "corroborate" that subregions of the PO and dpIns are specific to nociception/pain. The reported resting FC analyses do not directly corroborate this, and thus I suggest removing or changing this wording.

We agree and – in accordance with other remarks from reviewer #2 and #3, we completely rephrased this sentence, together with a large part of the Discussion.

- The first section of the Methods ("study design considerations") could be considerably shortened, as many of the details are not necessary for reproducing the present study.

According to this remark, the section was significantly shortened, and the focus was directed to the description of methods used in our study.

Reviewer #2

This is an elegant paper. The authors generate a pain sensitivity brain resting state network and validate the results in 2 studies performed in different institutions. These results are compelling and set a new level of rigor that researchers in the field should emulate. Technically the study is excellent. There are however important weaknesses that need attention and would render the study more solid and understandable.

We thank the reviewer for her/his overall positive evaluation regarding the quality and novelty of our study, and for the critical remarks that helped us to further improve our paper.

1) The groupings of brain regions with each other is unclear. In figure 3 why is VAN+BG+Th grouped together? Other groupings are based on RSNs, while some are purely anatomical, like the cerebellum. At least the reasoning of these parcellations needs clarity.

The groupings (or modules) were simply defined based on the “first level” of the multi-resolution MIST brain atlas (Urchs et al., 2017). This atlas is fully data driven (but spatially constrained) and was generated based on resting-state data of N=198 subjects by the BASC (bootstrap analysis of stable clusters) algorithm (Bellec et al., 2010). The MIST atlas is a multi-resolution atlas, with annotated functional parcellations at nine resolutions from 7 to 444 functional parcels. (Importantly, while the parcellation was not based on hierarchical clustering, there is a considerable overlap between the levels with different resolution.) While regional timeseries were obtained from the level with 122-regions, modules were defined based on the 7-region level. We considered obtaining more group labels, e.g. by separating the VAN+BG+Th module, but we decided to give preference to the data driven modules provided by the atlas. We clarified the definition of region groupings in the Methods.

2) The overall functional anatomical parcellation also needs further explanation. Perhaps illustrated in the supplement.

The details described above were indeed not fully reflected in the previous version of our manuscript. We have now added further details regarding this approach and the definition of the modules in the Methods section of the revised manuscript. Please see also our response to remark 3 of Reviewer #3.

3) It is also not clear how variable a given parcel may be from one subject to the next. As the functional networks are based on subject space anatomy, there must be distinct extents of variations for individual parcels. Please clarify and expound.

Thank you for giving us the opportunity to clarify the benefits of our atlas individualization method which is indeed an important step of our connectivity analysis pipeline.

First, masking of brain atlas regions with gray matter in the subject-space can be considered to be equivalent to a standard space masking (instead of the functional data, the region masks are transformed), and requires less interpolation steps on the functional data.

Second, atlas individualization was performed by masking with the individual gray-matter, this entails that regions are indeed different from subject to subject.

However, mainly due to co-registration inaccuracies, the conventional (standard-space) method will also result in different ROIs from subject to subject. (Although, ROIs look the same for each subject in the atlas-space, if we transform them back to individual space, the subject-to-subject differences will be easy to see.) Actually, inputting information from the tissue segmentation process by masking with individual gray matter is expected not to increase, but to decrease this variability, as some co-registration-inaccuracies will be also masked out. Namely, with our atlas-individualization, the final regional signal will originate with a high probability predominantly from gray matter voxels for each subject (which we carefully checked for all subjects), while with the conventional method, a variable ratio of gray- and white matter voxels are included for every subject.

To illustrate the variability stemming from subject-level anatomical differences (and captured by our atlas individualization method), we constructed study-specific “probability maps”, also used for Figure 3B. Example probability maps are now reported in the supplementary material (Figure S8). This new figure illustrates that the regional probability maps (and the individual regions) are not very different from a group-level gray-matter-based masking, which is a consequence of the high-precision image standardization procedure we used (Figure S2-4),

4) Although the authors repeatedly indicate that their signature is distinct and not directly comparable to pattern-based signatures for predicting presence of pain, they have a figure where these signatures are compared to their own. This is really a comparison between apples and oranges and simply leads to confusion. I strongly urge removing the figure and avoiding the comparisons. Activity and connectivity are different concepts and cannot be lumped together.

We agree with this point which has also been brought up by Reviewer #1 (comment 4). We have completely deleted Figure 4 and the related sentences from the manuscript.

5) A large part of the discussion again suffers from the temptation of lumping connectivity with activity. The specific brain regions and their link to pain perception are discussed although this concept is inconsistent with the network being the determining factor. The discussion of the role of S1 or S2, for example, with its specific connectivity is simply an over interpretation. The fact is that the identified network seems very distinct. For example it does not include posterior insulation, a region being debated to have some specificity for nociceptive information encoding in the cortex. In contrast, very large portions of the cerebellum seem involved in their signature and this is novel and perhaps uniquely linked with pain sensitivity.

We are thankful to the reviewer for raising this important point regarding connectivity vs. activity which has helped us to be more concise in our wording and interpretation in the revised version of the manuscript.

As suggested by the reviewer, we now put more emphasis on the potential dissimilarities of the predictive connectivity pattern identified in our study and the acknowledged brain activity pattern seen in the context of pain (also see point 6), such as the cerebellum and the lack of posterior insula (line 310). However, please note that, while we don't see posterior insula, the atlas region annotated as PO/STG might be very closely related. Namely, resting-state activity of these areas is known to be highly correlated (e.g. they belong to the same ROI in the “64-

region-level” of the MIST atlas) and the applied machine learning approach tends to pick only one from a set of correlated features. That might, in part, explain why the identified predictive connectivity signature is a relatively sparse network (a predictive node has in average 1.2 links). That also means that the reported predictive signature should be considered as a ‘sparse representation’ of the underlying true connectivity pattern. While due to the inherent variability in the feature selection procedure many different “sparse-signatures” may exist, our external validation procedure confirmed the predictive validity of the reported pattern and allows the interpretation of the single connections. We have clarified this issue in the revised discussion.

Moreover, throughout the discussion, we made it explicit whether we discuss key nodes of the predictive *connectivity* pattern in relation to brain *activity* or *connectivity*, as identified in a vast literature of pain neuroimaging.

6) A far simpler interpretation of the network is that it is a global reflection of how nociceptive information interacts with brain properties that determine the personality and cognitive and emotional and memory experiences of a given subject. Thus, searching in the literature for evidence that pieces of the network have some direct relationship with pain/nociception encoding regions of the brain remains unconvincing and perhaps points in the wrong direction.

We fully agree that these reverse-inference fueled discussions of single brain areas (or their connections) do actually artificially downscale the insights that are reflected in the connectivity signature as a whole and should be seen with caution.

We added more discussion on some general properties of the predictive network, e.g. its sparseness. We also significantly shortened the region-oriented, reverse inference-based discussion and sharpened it towards connectivity.

At line 335, now we summarise that (i) there is no brain area that is selectively and exclusively associated with pain, (ii) pain most probably emerges from the activity and connectivity of multiple brain areas that dynamically and flexibly connect to other networks subserving multiple functions, and (iii) that the network identified here represents this interaction of “pain-related” regions with brain properties that determine an individual’s cognitive, emotional and memory experiences related to pain.

Overall the study remains very exciting. However, the results are overinterpreted and compared to outcomes that are not relevant. The proposed changes clarifications should be easy to implement and would enhance the paper.

We are thankful for this encouraging evaluation and for the constructive recommendations. Thanks to these remarks, we believe we managed to significantly sharpen the discussion of the paper and eliminated overinterpretations.

Reviewer #3

This is a very interesting topic. My main concerns are as follows:

1. *The aim of the study is to using resting state functional connectivity to predict pain sensitivity. They used three pain threshold measurements in the manuscript. It would be helpful to demonstrate the association of these three measurements and discuss the reliability of pain sensitivity.*

Thank you for this important remark. We would like to clarify, that the dedicated aim of our study was to identify a brain pattern that is related to pain sensitivity in a more general, modality-independent way, as in (Zunhammer et al., 2016). However, we agree that demonstrating the association of these three measurements and how they are captured by the composite score is of crucial importance.

To this end, we have now added two supplementary figures (Figure S5 and S7) about the correlations between HPT, CPT and MPT and how they relate to the composite score. Figure S7 suggests that the composite score successfully captures variability in all three modalities. We clarified our motivations in the introduction, accordingly.

Please also see our replay to the first comment from Reviewer #1, which has a very similar topic.

2. *The authors used a single composite measure by averaging HPT, CPT, and MPT. Because heat pain and mechanical pain activate different fibers, it is likely that brain patterns encoding HP and MP are different. To generalize the results, the authors may want to 1) compare the predictive patterns for different pain modalities and/or 2) build the model from one modality and test on other modalities.*

We certainly agree, that training and comparing models for HPT, CPT and MPT would substantially aid our understanding of how neural processing differs in pain modalities and how the general, modality-independent trait of pain sensitivity is established. Given that such an analysis is expected to provide several new insights and would require an extensive discussion, we feel that it is out of the scope of the present study and provides an excellent topic for further research.

Notably, our decision to search for a composite, largely modality-independent signature has the advantage of mitigating modality-specific confounds (like reaction times which confound the thermal thresholds but not the mechanical thresholds) and is therefore a straightforward first step before mapping predictive patterns across modalities.

The former version of the manuscript, and especially part D of Figure 2 may have distracted the reader from this overall goal. Therefore, we deleted part D from Fig.2. and changed the caption and the text (line 122) accordingly and added a clarification (line 43 and 63) of our motivations into the Introduction and the start of the Results.

3. *The rationale of using MIST for brain parcellation is unclear. Since MIST is a very new atlas, I wonder if the authors have validated their findings based on other well-established brain parcellation strategies (e.g., Yeo atlas, Power atlas).*

We are thankful helping us improve the manuscript by clarifying the rationale beyond our choice of brain atlas which, together with the applied atlas individualization approach (see also point 3 by Reviewer #2), is indeed an important step in our connectivity analysis pipeline.

The choice of brain parcellation was motivated as follows:

1) We avoided using study-specific data driven parcellations (e.g. ICA, k-mean, etc.) as those incorporate information about the pooled sample and might be problematic during cross-validation (parcel information from the test set leaks to the training set).

2) In the recent study of Dadi and colleagues (Dadi et al., 2019), among all pre-defined atlases, the one called ‘BASC’ was found to perform best. Please note that the MIST atlas (Urchs et al., 2017) is based on the BASC (bootstrap analysis of stable clusters) algorithm (Bellec et al., 2010) and therefore simply called as BASC in the Dadi-paper.

We selected the MIST atlas based on its superior performance in this study (as compared to the AAL, Power and the Harvard-Oxford atlases)

We also considered the use of the Yeo and Glasser atlases, but these do not cover the cerebellum.

4. *Line 786. How did the authors pre-select features? Ref 42 may not be the one to cite. If K was selected from 10 to 200, this number is \ll original feature space ($k= 7503$), and if the optimal K was 25, which is smaller than N ($N1 = 35$), I would question the necessity of using elastic net. A recent benchmark study by Dadi and colleagues [Neuroimage, 192 (2019) 115-134] demonstrated that non-sparse linear methods outperformed sparse linear methods in connectome-based prediction tasks. The authors need to clarify the feature selection procedure.*

Thank you for raising this important issue; as highlighted also by recent scientific discussion (Dadi et al., 2019; Spisak et al., 2019), the a-priory analysis decisions are of crucial importance in predictive modelling.

The use of elastic net was a decision drawn prior to the analysis (that is, before knowing the “optimal” number of features). Our main motivation to choose elastic net was that it allows to optimize sparsity (L1 vs. L2 regularization) as a hyperparameter. We believe that one should be careful when making assumptions about sparsity of the discriminative signal before the analysis. For more information, please refer to the preprint of our recent commentary (Spisak et al., 2019) on the benchmarking paper by (Dadi et al., 2019), where we highlight some potential drawbacks of using non-sparse regularization as a default technique.

Pre-selection of features is also a related topic (see our preprint). It was performed by selecting the K “best” features with strongest relationships to the target variable, based on an F-score from multiple regressions (as implemented in ‘scikitlearn’).

We have revised this section in the Methods to clarify the above details and fixed reference 42.

5. *The details of external validation are missing. Did the authors take dot product between connectivity patterns and non-zero feature weights?*

Yes, we did a dot product, after applying the feature transformation obtained during training (practically a scaling, but based on the summary statistics of Sample 1). This very important information was, indeed, missing in the previous version of the text.

6. *Statistical significance for MAE is not provided. The authors may consider permutation testing to assess the performance of MAE.*

We agree that permutation test-based p-values are important when assessing the goodness of prediction. We performed permutation test to assign p-values to all error measures and extended the text and the supplementary material (Table S1) accordingly.

7. *Results in Figure 2D suggest that the RPN may fail to generalize to single modalities (e.g., mechanical pain in the upper panel and heat pain in the lower panel). Those results may be largely affected by the collinearity between the composite scores and single-modality scores.*

The observation (also raised in the first comment of Reviewer #1) that the correlation between HPT and the RPN-score differs in Study 2 and 3 is correct, however please note that the error margin for this correlation in study 3 (which had the lowest N) was so wide that no conclusions should be drawn (e.g. a negative correlation is easily within the error interval off). As Figure 2D was confusing in other aspects, as well (noted by all three reviewers) and distracting the focus from the primary outcome, we removed this part of the figure. For a better characterization of the pain thresholds and the composite score, please see supplementary figure S5 and supplementary table S7 and our replay to the first comment of reviewer #1 for more detail.

8. *The importance of the connectivity in the prediction was represented by the raw coefficient weights but not statistical values. Statistical assessments for those coefficients are preferred. Details can be found in Wager's NPS in Figure 1 NEJM 2013.*

We agree that bootstrapping based p-values can provide additional information about the robustness and stability of the single coefficients provided by the training procedure.

However, we believe that assigning p-values to each feature is much more relevant in the case of machine learning pipelines *without intrinsic feature selection*. The NPS falls into this category, as it was trained on PCA components which used (a varying amount of) information from *all voxels* in the brain. The bootstrapping analysis in the NPS-paper to select the most predictive voxels was thus an elemental requirement for visualization and interpretation purposes.

In contrast, our method involves intrinsic feature selection, thus no post-hoc statistical analysis is needed to identify a lower, “interpretable” number of important features.

Of course, bootstrapped p-values and confidence intervals can still provide extra information on the reliability of coefficients for these models, too.

However, performing an exact, “all-inclusive” post-selection inference for predictive models with feature-selection (as in our case) is not trivial for two reasons: (i) there is a pessimistic bias due to the regularization approach, which pushes coefficients towards zero, and (ii) an optimistic bias since the variables that are selected will tend to be those that are significant. While exact post-selection inference is subject of ongoing research (Lee et al., 2016), it is also possible to perform a much simpler “conditional coverage” analysis, where p-values are conditioned on the selection event. Note however, that the interpretation of these conditioned p-values is limited.

Despite these methodological considerations, based on the reviewer's suggestion, we have now performed a "conditional coverage" bootstrapping analysis (see Methods for details). It revealed that p-values (conditional on feature selection) are below 0.05 for the majority of the predictive connections. Exceptions are the five smallest coefficients. While, of course, the lack of statistical significance (at an arbitrary alpha level of $p=0.05$) for these connections does not imply the lack of predictive effect, the point-estimates of the corresponding coefficients – and accordingly, their contribution to the predicted score - are very small, nevertheless.

We added these results into Supplementary Table S4 (but not in the main text due to the easy mis-interpretation of the conditional p-values) and tuned down the discussion of connections with a high ($p>0.05$) post-selection p-value / low predictive weight. Additionally, we now discuss in the manuscript, that our RPN-signature is not an "ultimate neural signature" of pain sensitivity, but rather a "sparse" representation of the underlying true connectivity pattern. While potential training sample-related instabilities in feature selection might result in different "sparse-signatures", our external validation procedure ensures the predictive validity of the reported pattern.

9. *Line 322. It is overstated that the RPN is specific to pain sensitivity and is not driven by the general sensitivity to sensory stimuli. In this study, other sensory modalities have not been tested.*

We agree and added a relevant discussion in the limitations section and eliminated the overstatements. Note however that, for the investigated sensory modalities (heat, cold and mechanical pain) the specificity of the RPN-signature to pain sensitivity is strongly supported by our results that it is not correlated with the sensory detection thresholds (CDT, WDT and MDT in Table 1).

Minor:

1. *The authors should check all their cited references. A lot of references do not match the information they were trying to provide. Two examples: Line 63, Ref 8. Line 289, Ref 33.*

Thank you for the detailed review, we carefully checked all references and fixed the broken ones. In general, we believe that all the remarks helped us improve the manuscript and to provide a concise and comprehensive presentation of our study.

List of most important changes in the manuscript

Introduction

- clarification of study motivations (line 45)

Results

- clarification of study motivations (line 65)
- early clarification that pain sensitivity was based on a composite measure of heat, cold, and mechanical thresholds.
- Added permutation test-based p-values for the error metrics
- Added bootstrapping-based confidence intervals and p-values for the coefficients of the predictive model (Table 2)
- deleted part D from Fig.2.
- completely deleted Fig.4.

Discussion

- tuned down overstatements regarding clinical relevance
- sharpened discussion by:
 - o explicitly differentiate between *activity and connectivity*
 - o eliminated localization-based overinterpretations of the RPN-network-nodes
 - o put more emphasis on the potential dissimilarities between the predictive connectivity pattern identified in our study and the acknowledged brain activity patterns seen in the context of pain
 - o introducing the interpretation that the network identified here represents an interaction of “pain-related” regions with brain properties that determine an individual’s cognitive, emotional and memory experiences related to pain.

Methods

- shortened the discussion of general methodological considerations
- added several details about our choice of brain atlas and the individualization of brain atlas regions
- clarified our a-priori analysis decisions regarding predictive modelling
- added more detail about the external validation procedure

Supplementary material

- added Table S4 with bootstrapping-based p-values and confidence intervals.
- added Figure S5 with the correlations between HPT, CPT and MPT in all cohorts.
- added Figure S6 about the relation between the composite pain sensitivity score and its subscales, HPT, CPT and MPT.
- added Figure S7, an illustration of the brain atlas individualization procedure.

Full manuscript

- changed “susceptibility to pain” to “sensitivity to pain”
- removed redundant parts (e.g. hyperlinks)
- avoided the use of the word “direct”, when referring to partial correlation-based functional connectivity
- added several important new references (Cheng et al., 2017, 2015; Lee et al., 2016; Riedl et al., 2011; Spisak et al., 2019)
- checked all references and fixed the broken ones.

References

- Bellec, P., Rosa-Neto, P., Lyttelton, O.C., Benali, H., Evans, A.C., 2010. Multi-level bootstrap analysis of stable clusters in resting-state fMRI. *Neuroimage* 51, 1126–1139. <https://doi.org/10.1016/j.neuroimage.2010.02.082>
- Cheng, J.C., Bosma, R.L., Hemington, K.S., Kucyi, A., Lindquist, M.A., Davis, K.D., 2017. Slow-5 dynamic functional connectivity reflects the capacity to sustain cognitive performance during pain. *Neuroimage* 157, 61–68. <https://doi.org/10.1016/j.neuroimage.2017.06.005>
- Cheng, J.C., Erpelding, N., Kucyi, A., DeSouza, D.D., Davis, K.D., 2015. Individual Differences in Temporal Summation of Pain Reflect Pronociceptive and Antinociceptive Brain Structure and Function. *J. Neurosci.* 35, 9689–9700. <https://doi.org/10.1523/jneurosci.5039-14.2015>
- Dadi, K., Rahim, M., Abraham, A., Chyzyk, D., Milham, M., Thirion, B., Varoquaux, G., 2019. Benchmarking functional connectome-based predictive models for resting-state fMRI. *Neuroimage* in press. <https://doi.org/10.1016/J.NEUROIMAGE.2019.02.062>
- Lee, J.D., Sun, D.L., Sun, Y., Taylor, J.E., others, 2016. Exact post-selection inference, with application to the lasso. *Ann. Stat.* 44, 907–927.
- Riedl, V., Valet, M., Wöller, A., Sorg, C., Vogel, D., Sprenger, T., Boecker, H., Wohlschläger, A.M., Tölle, T.R., 2011. Repeated pain induces adaptations of intrinsic brain activity to reflect past and predict future pain. *Neuroimage* 57, 206–213. <https://doi.org/10.1016/j.neuroimage.2011.04.011>
- Spisak, T., Kincses, B., Bingel, U., 2019. Optimal choice of parameters in functional connectome-based predictive modelling might be biased by motion : comment on Dadi et al . *bioRxiv* 0–4.
- Urchs, S., Armoza, J., Benhajali, Y., St-Aubin, J., Orban, P., Bellec, P., 2017. MIST: A multi-resolution parcellation of functional brain networks. *MNI Open Res.* 1, 3. <https://doi.org/10.12688/mniopenres.12767.1>
- Zunhammer, M., Schweizer, L.M., Witte, V., Harris, R.E., Bingel, U., Schmidt-Wilcke, T., 2016. Combined glutamate and glutamine levels in pain-processing brain regions are associated with individual pain sensitivity. *Pain* 157, 2248–2256. <https://doi.org/10.1097/j.pain.0000000000000634>

Reviewers' comments:

Reviewer #1 (Remarks to the Author):

The authors have successfully addressed most of my concerns. I think there is still a remaining issue with the reporting of HPT vs. CPT vs. MPT results. While I understand that the authors are choosing to focus mainly on the composite score, the issue of possible divergence among distinct measures of pain thresholds needs more attention, given that different sensory pathways mediate these distinct modalities.

In Figure S5, the plots do not have x and y labels, and it is difficult to decipher what is being shown. Also, given the results, the figure legend should not read "The predicted pain sensitivity score was not only correlated with the observed QST-based composite pain sensitivity score but also with its modality-specific components, the cold, heat and mechanical pain thresholds." As noted by the authors in their reply, these correlations were not consistent.

I am also not sure how to interpret Figure S6, given that it doesn't appear that statistics were performed. It really just looks like a mixed bag of subjects, where some show increased and others show decreased composite scores relative to modality-specific thresholds. What is the take-away message supposed to be? Are there any specific relationships between modality-specific thresholds and the composite scores?

I recommend further clarification as well as additions to the discussion about the associations among HPT vs. CPT vs. MPT and the potential meaning of the composite score.

Reviewer #2 (Remarks to the Author):

The authors have properly responded to earlier comments. This manuscript is of very high quality and demonstrates a robust approach for performing unbiased research. Moreover, it shows that brain properties can identify a validated pain-sensitivity trait.

A. Vania Apkarian

Reviewer #3 (Remarks to the Author):

The authors addressed some of my concerns; however, some concerns remain:

1. Is the method to summarize different pain thresholds a well-accepted method? The authors cited one publication from their group, but I am not sure how widely accepted the single composite is. One concern is if a new pain threshold measurement is added, the pain sensitivity score will change, as may the network.
2. The authors provide Sup figure 5 to show the association between three thresholds. It seems that at least some of the three pain thresholds are only weakly associated. The lack of p values makes it difficult to judge the significance of these correlations. Sup Figure 6 (not figure 7) is helpful, but a formal statistical analysis would be better.
3. Although 'BASC' was found to perform best among all pre-defined atlases in Dadi and colleagues (Dadi et al., 2019), Dadi et al. used different disorders in their data analysis, and pain thresholds / sensitivity was not included in the analysis. Thus, it is necessary to use another atlas to validate the findings obtained from one atlas. If the author is worried that Yeo atlas does not include cerebellum, AAL or Harvard or Oxford atlases could be considered.
4. There is a recent publication using connectome to predict pain threshold (Tu et al., Neuroimage 2019). The author may want to compare their finding with the results from the recent publication.

Point-by-point response to reviewers' comments

Reviewer #1

The authors have successfully addressed most of my concerns. I think there is still a remaining issue with the reporting of HPT vs. CPT vs. MPT results. While I understand that the authors are choosing to focus mainly on the composite score, the issue of possible divergence among distinct measures of pain thresholds needs more attention, given that different sensory pathways mediate these distinct modalities.

We are thankful for this remark and agree that the internal consistency of the distinct measures of pain thresholds has important implications regarding the neuroscientific validity of the composite pain sensitivity score we used and the interpretation and generalizability of the proposed brain signature of pain sensitivity. We targeted this question with a multi-stage analysis (*Supplementary Analysis 1*).

In Figure *SA1.1*, we now present a more detailed reporting of the divergence across pain threshold measures. We have found that two out of the three possible correlations between the single pain thresholds are statistically significant ($\text{cor}(\text{HPT}, \text{CPT}) = -0.51$, $p < 0.0001$; $\text{cor}(\text{HPT}, \text{MPT}) = 0.19$, $p = 0.03$ and $\text{cor}(\text{CPT}, \text{MPT}) = -0.06$, $p = 0.26$). The fact, that thermal thresholds show a stronger association to each other than to the mechanical threshold, is in line with the partially different, “modality-specific” pathways mediating pain in these measures (Rolke et al., 2006).

Moreover, the experimental procedures for HPT and CPT are more similar compared to MPT which may explain some of the shared variance. For instance, procedures for the thermal thresholds differ only in the direction of temperature changes and in the wording of instructions; both HPT and CPT are obtained with an automated thermode and based on a method of ascending limits. In contrast, the MPT is obtained with hand-held pin-pricks, based on a staircase method, and involves more stimulus repetitions.

Importantly, our analysis (especially *Supplementary Analysis 1 - Q1, Step 3*) corroborates previous results (Neddermeyer et al., 2008) providing evidence for a pain-sensitivity component that is shared across distinct measures of pain thresholds. (For a discussion of how well this shared component is captured by the predictive model, see our response to Reviewer #3). We have added a more detailed discussion of our results regarding the internal consistency among distinct measures of pain thresholds in the main text (line 297).

In Figure S5, the plots do not have x and y labels, and it is difficult to decipher what is being shown. Also, given the results, the figure legend should not read "The predicted pain sensitivity score was not only correlated with the observed QST-based composite pain sensitivity score but also with its modality-specific components, the cold, heat and mechanical pain thresholds." As noted by the authors in their reply, these correlations were not consistent.

We agree that *Supplementary Figure S5* of the previous revision did not provide sufficient detail regarding the relationship between HPT, CPT and MPT. We apologize for the missing information and for the error in the figure legend which was accidentally copy-pasted from another figure. In the revised manuscript, *Figure S5* is replaced by *Supplementary Analysis 1* and, specifically, *Figure SA1*. We ensured that *Figure SA1* contains all the requested information. Moreover, *Supplementary Analysis 1* provides additional insights regarding the internal consistency of the distinct pain thresholds.

I am also not sure how to interpret Figure S6, given that it doesn't appear that statistics were performed. It really just looks like a mixed bag of subjects, where some show increased and others show decreased composite scores relative to modality-specific thresholds. What is the take-away message supposed to be? Are there any specific relationships between modality-specific thresholds and the composite scores?

This figure originally aimed to aid the interpretation of the “arbitrary units” of the composite pain sensitivity score. However, it was indeed unintuitive for presenting any relationship between the variables. Similarly to *Figure S5*, *Figure S6* is now replaced by *Supplementary Analysis 1* (new Figures *SA1*, *SA2* and *SA4*), which clarifies that the relationship between modality-specific thresholds and the composite score is driven by a shared “modality-independent” component.

I recommend further clarification as well as additions to the discussion about the associations among HPT vs. CPT vs. MPT and the potential meaning of the composite score.

Based on the results of *Supplementary Analysis 1* and as outlined in our previous answer, we have now revised the discussion in the main text. Specifically, we now discuss that:

1. The observed moderate internal consistency and the specific correlation structure across the distinct pain thresholds have straightforward experiment-specific and neurobiological interpretations. See line 297 in the main text and the first paragraph of part *Q1* in *Supplementary Analysis 1*.
2. The validity of the composite score of (Zunhammer et al., 2016) as a prediction target is supported by the results of *Supplementary Analysis 1, Q1*, which demonstrates the internal consistency and the existence of a component shared across the distinct measures of pain threshold, to which we refer to as the “modality-independent” component (line 302 in the main text).
3. As shown in parts *Q2* and *Q3* of *Supplementary Analysis 1*, both the composite score of (Zunhammer et al., 2016) and the prediction based on it capture this “modality-independent” component of pain sensitivity. (line 306 in the main text).
4. The predictive model does not introduce any significant bias towards/against any of the modalities, as shown and discussed in part *Q4* of *Supplementary Analysis 1* and highlighted at line 309 in the main text.

Reviewer #2

The authors have properly responded to earlier comments. This manuscript is of very high quality and demonstrates a robust approach for performing unbiased research. Moreover, it shows that brain properties can identify a validated pain-sensitivity trait.

A. Vania Apkarian

Thank you. It is an honor to receive this comment.

Reviewer #3

The authors addressed some of my concerns; however, some concerns remain: 1. Is the method to summarize different pain thresholds a well-accepted method? The authors cited one publication from their group, but I am not sure how widely accepted the single composite is. One concern is if a new pain threshold measurement is added, the pain sensitivity score will change, as may the network.

We are thankful to *Reviewer #3* for raising this question as we agree that the appropriateness of the composite score of (Zunhammer et al., 2016) as a prediction target is a cornerstone of our study.

While the specific composite score we used in this study (introduced in Zunhammer et al., 2016) was not yet used by other groups, our new *Supplementary Analysis 1* strongly supports that it is a useful measure of the general sensitivity to pain and that it is an appropriate choice as prediction target, when focusing on the shared “modality-independent” aspects of pain sensitivity (as also described by e.g. Neddermeyer et al., 2008). Please also see our answers to *Reviewer #1* regarding the internal consistency of the pain thresholds.

To address the question of the “stability” of our prediction to the exact definition of the composite pain sensitivity score and, specifically, to the **effect of adding/removing a modality to/from** the composite score, we performed “leave-one-modality-out” and “leave-two-modalities-out” analyses (part *Q3* of *Supplementary Analysis 1*). These analyses revealed that the RPN-score displays a considerable **robustness** to the definition of pain sensitivity. Namely, while the correlation of the RPN-score with the original composite pain sensitivity score was 0.48, it ranged between 0.4 and 0.45 for three the possible leave-one-modality scores and between 0.24 and 0.44 for the leave-two-modalities-out scores (i.e. the single measures pain thresholds). All of these correlations were statistically significant ($p < 0.05$, see **Supplementary Analysis 1, part Q3** for more details).

Although, these analyses cannot be trivially extended to the case of adding a new, fourth pain threshold measurement, we believe that the RPN-signature remains valid for other pain thresholds, as long as they reflect the shared, “modality-independent” component. Based on reported cross-modality correlations, this is most probably the case e.g. for pressure, ischemic or electrical pain thresholds (Clark et al., 1956, Hastie et al., 2005, Bhalang et al., 2005, Neddermeyer et al., 2007, Tu et al., 2019).

We have revised our discussion of the composite score in the main text (see our response to the last comment of *Reviewer #1*), including the discussion of the robustness of the predictive performance of the RPN-signature to the choice of sensory modalities.

2. The authors provide Sup figure 5 to show the association between three thresholds. It seems that at least some of the three pain thresholds are only weakly associated. The lack of p values makes it difficult to judge the significance of these correlations. Sup Figure 6 (not figure 7) is helpful, but a formal statistical analysis would be better.

We apologize for the missing information. We have added the detailed statistics regarding the association of the different pain thresholds. Two out of the three possible correlations between the single pain thresholds were found to be statistically significant ($\text{cor}(\text{HPT}, \text{CPT}) = -0.51$, $p < 0.0001$; $\text{cor}(\text{HPT}, \text{MPT}) = 0.19$, $p = 0.03$ and $\text{cor}(\text{CPT}, \text{MPT}) = -0.06$, $p = 0.26$). The observed correlation structure has straightforward experiment-specific and neurobiological interpretations (Please see our response the first remark from *Reviewer #1*.) This and other information that was missing from the old *Fig. S5*, is now presented in *Fig. SAI* of *Supplementary Analysis 1*.

Moreover, the message of *Figure SA1* is now extended with additional analyses and figures. Specifically, *Supplementary Analysis 1* (specifically, part Q1, Step3) shows that all three modalities contribute significantly to the first principal component of the dataset. This suggests that, despite the partially different sensory pathways mediating the different pain thresholds, a shared, “modality-independent” component of pain sensitivity shapes pain thresholds in all three pain modalities, which is in line with previous investigations (Neddermeyer et al., 2008).

3. Although ‘BASC’ was found to perform best among all pre-defined atlases in Dadi and colleagues (Dadi et al., 2019), Dadi et al. used different disorders in their data analysis, and pain thresholds / sensitivity was not included in the analysis. Thus, it is necessary to use another atlas to validate the findings obtained from one atlas. If the author is worried that Yeo atlas does not include cerebellum, AAL or Harvard or Oxford atlases could be considered.

As the input features for the predictive model are timeseries averaged within atlas-based regions, the parcellation scheme of the atlas might, indeed, have an important effect on predictive modelling.

In *Supplementary Analysis 2*, we have performed a “worst-case” evaluation of the stability of the RPN-signature to possible parcellation-related inaccuracies/errors. This analysis, next to addressing the reviewer’s concern regarding the robustness of the reported signature to parcellation schemes, gives a **general lower-bound approximation of the stability of the RPN-signature to possible parcellation-related effects.**

First, a suboptimal parcellation (due to e.g. inaccurate parcellation scheme but, also, co-registration inaccuracies and low data quality) results in “mixed-signal” effects, i.e. the discriminative signal (“signal-of-interest”) is mixed with signal of other origin in the delineated region (either noise or neural signal from neighboring areas; “signal-of-no-interest”). To evaluate the effect of mixed-signal within the regional timeseries, we added (various amounts of) noise to all regional timeseries. Note, that - as signal in neighboring brain regions is usually correlated - adding uncorrelated noise is a worst-case scenario and provides an **upper bound on the detrimental effect of mixed signals.**

The results reveal that the RPN-signature displays a remarkable robustness to “mixed signal” effects (*Supplementary Analysis 2, Part 1*) and the prediction remains - on average - significant even if about three quarter (noise weight=3 on **Figure SA2.1**) of the area of the region contains exclusively “signal-of-no-interest”.

Second, depending on parcellation (brain coverage of atlas) or data quality (susceptibility artefact, sequence field-of-view), some regions may partly or entirely “fall out” from the feature set (“drop-out” effect). Therefore, we tested, how randomly zeroing out the regional timeseries in a given number of regions affects prediction accuracy. We have found that the RPN-signature displays a remarkable stability to the drop-out of regions and the prediction remained significant, on average, when dropping out 19 out of the 21 predictive regions, although - as expected - prediction accuracy significantly decreased with an increasing number of dropped regions. (*Supplementary Analysis 2, Part 2*).

These new results are in line with (Dadi et al, 2019, especially Figure 5) in suggesting that brain parcellation - at least as long as well-established atlases are used - has only a slight effect on the accuracy of predictive models (AUC difference less than 0.05). We added a related discussion at line 193 in the main text.

4. There is a recent publication using connectome to predict pain threshold (Tu et al., Neuroimage 2019). The author may want to compare their finding with the results from the recent publication.

We are thankful to *Reviewer #3* for this great reference which was published after the submission of our manuscript to Nature Communications. We have integrated this reference in the main text of our manuscript (lines 22, 297, 323, 345 and 363).

Importantly, this paper supports the results of our new analysis (*Supplementary Analysis 1*) of the internal consistency across pain thresholds, as Tu et al. report a significant correlation of $R=0.41-0.74$ ($p<0.05$ at their sample size of $N=24$) between mechanical (pressure pain) and thermal (heat pain) thresholds.

Despite this significant correlation between the pain threshold measures, Tu et al. chose to analyze pain modalities separately. Therefore, their results cannot be directly compared to ours, but rather extend it, allowing for the deduction of hypotheses regarding the differences between the modality-specific and modality-independent components of pain sensitivity.

Importantly, the predictive performance of $R=0.5-0.6$ demonstrated by Tu et al. is comparable to our study. As the modality-independent component of pain sensitivity we aim to predict is obviously of smaller magnitude than the sum of it and the modality-specific component, the predictive performance of the RPN-signature is remarkable and suggests that using the composite score of Zunhammer et al. as prediction target allowed us to model a relevant amount of variance in the general sensitivity to pain across individuals.

List of changes in the manuscript

1. Added *Supplementary Analyses 1* and *2* as pdf and also available at the following links:
https://raw.githubusercontent.com/spisakt/RPN-signature/master/notebooks/Supplementary_Analysis_1.html
https://raw.githubusercontent.com/spisakt/RPN-signature/master/notebooks/Supplementary_Analysis_2.html
Alternatively, to edit and run the code:
https://nbviewer.jupyter.org/github/pni-lab/RPN-signature/blob/master/notebooks/Supplementary_Analysis_1.ipynb
https://nbviewer.jupyter.org/github/pni-lab/RPN-signature/blob/master/notebooks/Supplementary_Analysis_2.ipynb
By clicking on “Execute on Binder” in the top right corner, the reviewers can enter the interactive mode where the code of the analysis can be edited and run in a dedicated python environment.
2. Referring to and discussing the results of *Supplementary Analyses 1* and *2* at lines 66, 124, 190 and 297.
3. The old supplementary figures *S5* and *S6* is now part of our new *Supplementary Analysis 1* (figures *SA1.1*, *SA1.2* and *SA1.4*). Moreover, as detailed in our point-by-point responses below, the message of the old figures *S5* and *S6* is significantly extended by *Supplementary Analysis 1*.
4. Added and discussed a new reference (Tu et al., 2019) at lines 22, 297, 326, 349 and 367.

References

- (Bhalang et al., 2005)** Bhalang K, Sigurdsson A, Slade GD, Maixner W. Associations among four modalities of experimental pain in women. *The Journal of Pain*. 2005 Sep 1;6(9):604-11.
- (Clark et al., 1956)** Clark JW, Bindra D. Individual differences in pain thresholds. *Canadian Journal of Psychology/Revue canadienne de psychologie*. 1956 Jun;10(2):69.
- (Craddock et al., 2012)** Craddock RC, James GA, Holtzheimer III PE, Hu XP, Mayberg HS. A whole brain fMRI atlas generated via spatially constrained spectral clustering. *Human brain mapping*. 2012 Aug;33(8):1914-28.
- (Hastie et al., 2005)** Hastie BA, Riley III JL, Robinson ME, Glover T, Campbell CM, Staud R, Fillingim RB. Cluster analysis of multiple experimental pain modalities. *Pain*. 2005 Aug 1;116(3):227-37.
- (Neddermeyer et al., 2008)** Neddermeyer, Till J., Karin Flühr, and Jörn Lötsch. "Principle components analysis of pain thresholds to thermal, electrical, and mechanical stimuli suggests a predominant common source of variance." *Pain* 138.2 (2008): 286-291.
- (Rolke et al., 2006)** Rolke R, Baron R, Maier CA, Tölle TR, Treede RD, Beyer A, Binder A, Birbaumer N, Birklein F, Bötefür IC, Braune S. Quantitative sensory testing in the German Research Network on Neuropathic Pain (DFNS): standardized protocol and reference values. *Pain*. 2006 Aug 1;123(3):231-43.
- (Zunhammer et al., 2016)** Zunhammer M, Schweizer LM, Witte V, Harris RE, Bingel U, Schmidt-Wilcke T. Combined glutamate and glutamine levels in pain-processing brain regions are associated with individual pain sensitivity. *Pain*. 2016 Oct 1;157(10):2248-56.

REVIEWERS' COMMENTS:

Reviewer #1 (Remarks to the Author):

The authors have successfully addressed all of my concerns, and I am convinced by their detailed supplementary analyses.

Reviewer #3 (Remarks to the Author):

The authors have addressed all my comments / concerns. I have no further question / comment.